# Paired immunoglobulin-like receptor B is an entry receptor for mammalian orthoreovirus

Pengcheng Shang [1,2], Joshua D. Simpson [3], Gwen M. Taylor [1,2], Danica M. Sutherland [1,2], Olivia L. Welsh[1,2], Pavithra Aravamudhan[1,2], Rita Dos Santos Natividade [3], Kristina Schwab[4], Joshua J. Michel[1], Amanda C. Poholek [1,4], Yijen Wu [5], Dhivyaa Rajasundaram[1], Melanie Koehler[6], David Alsteens [3,7] & Terence S. Dermody [1,2,8] ✉

Mammalian orthoreovirus (reovirus) infects most mammals and is associated with celiac disease in humans. In mice, reovirus infects the intestine and disseminates systemically to cause serotype-specific patterns of disease in the brain. To identify receptors conferring reovirus serotype-dependent neuro-pathogenesis, we conducted a genome-wide CRISPRa screen and identified paired immunoglobulin-like receptor B (PirB) as a receptor candidate. Ectopic expression of PirB allowed reovirus binding and infection. PirB extracelluar D3D4 region is required for reovirus attachment and infectivity. Reovirus binds to PirB with nM affinity as determined by single molecule force spectroscopy. Efficient reovirus endocytosis requires PirB signaling motifs. In inoculated mice, PirB is required for maximal replication in the brain and full neuropathogenicity of neurotropic serotype 3 (T3) reovirus. In primary cortical neurons, PirB expression contributes to T3 reovirus infectivity. Thus, PirB is an entry receptor for reovirus and contributes to T3 reovirus replication and pathogenesis in the murine brain.

As obligate intracellular microbes, viruses depend on numerous cellular factors to replicate. Viruses initiate infection by penetrating through cell membranes to enter the cytoplasm, which is often a multistep process requiring interactions with cell-surface attachment factors and internalization receptors[1–5]. Engagement of attachment factors and internalization receptors often dictates host range, transmission route, cell and tissue tropism, and disease severity[3,6–8]. Viruses have evolved diverse strategies to interact with receptors, which can be dependent on virus strain[8–11] or cell type[12]. Virus-uptake mechanisms also can be dictated by interactions of different virion components with multiple receptors[7,12–16]. However, mechanisms governing virus-receptor interactions and functions of receptors in viral

pathogenesis are not well understood for many pathogenic viruses. Mammalian orthoreovirus (reovirus) is a generalist pathogen with a broad mammalian host range[17]. Reovirus-receptor interactions are viral serotype- and cell type-specific and involve several capsid protein-receptor pairs[18]. Therefore, reovirus serves as a tractable experimental model to investigate how the highly orchestrated process of viral receptor binding and internalization influences viral tropism and disease.

There are three reovirus serotypes (T1, T2, and T3), of which, T1 and T3 have been most thoroughly characterized[17,18]. Following fecal-oral transmission, reovirus establishes primary replication in the intestine and disseminates systematically to secondary sites including

---

[1]Department of Pediatrics, University of Pittsburgh School of Medicine, Pittsburgh, PA, USA. [2]Institute of Infection, Inflammation, and Immunity, UPMC Children's Hospital of Pittsburgh, Pittsburgh, PA, USA. [3]Louvain Institute of Biomolecular Science and Technology, Université catholique de Louvain, Louvain-la-Neuve, Belgium. [4]Department of Immunology, University of Pittsburgh School of Medicine, Pittsburgh, PA, USA. [5]Department of Developmental Biology, University of Pittsburgh School of Medicine, Pittsburgh, PA, USA. [6]Leibniz Institute for Food Systems Biology at the Technical University Munich, Freising, Germany. [7]WELBIO Department, WEL Research Institute, Wavre, Belgium. [8]Department of Microbiology and Molecular Genetics, University of Pittsburgh School of Medicine, Pittsburgh, PA, USA. ✉e-mail: terence.dermody@chp.edu

the central nerve system (CNS). The dissemination route from intestine to the CNS and tropism for specific CNS cells is serotype-dependent. In newborn mice, T1 reovirus spreads hematogenously to infect ependymal cells and cause hydrocephalus. In contrast, T3 reovirus spreads to the CNS using both hematogenous and neural routes to infect neurons and cause lethal encephalitis. Serotype-specific dissemination pathways and CNS tropism are dictated by the trimeric σ1 viral attachment protein, likely by engaging cell type-specific receptors.

Reovirus attachment and internalization are coordinated by interactions of three different viral capsid proteins with several host factors[19–24]. Reovirus σ1 protein binds to sialic acid (SA) with low affinity as an attachment factor[21,25,26]. The σ1 protein also binds to junctional adhesion molecule A (JAM-A) with much higher affinity at a different interface in the protein[23,24,27,28]. The pentameric λ2 protein, which anchors the σ1 protein into the virion[17], promotes viral internalization by binding β1 integrins[20,29,30]. Outer-capsid protein σ3 engages the human homolog of the Nogo-66 receptor 1 (hNgR1)[19,22]. However, our knowledge of reovirus receptor use in vivo is limited. SA binding contributes to virus dissemination but not to tropism[31,32]. JAM-A is essential for hematogenous but not neural dissemination and is dispensable for reovirus replication in the CNS[33]. Likewise, NgR1 is not required for reovirus CNS replication or disease in mice[34]. Therefore, the known reovirus attachment factors and internalization receptors do not fully explain reovirus serotype-specific CNS tropism and disease.

To fill this knowledge gap, we used a gain-of-function CRISPR activation (CRISPRa) screen. Paired immunoglobulin-like receptor B (PirB), which is a member of the leukocyte immunoglobulin-like receptor (LILR) family, was identified as a top candidate. PirB is expressed on immune cells[35–37] and neurons[38] and serves as a receptor for major histocompatibility complex class I (MHC-I) proteins and myelin-associated inhibitors (MAIs)[35,38–40]. Ectopic expression of PirB in non-susceptible cells promotes reovirus binding and infection, and reovirus directly interacts with PirB with high affinity. Alteration of signaling motifs in the PirB cytoplasmic tail compromises reovirus entry. Yields of T3 reovirus, but not T1 reovirus, are diminished in the brains of PirB[−/−] mice relative to those in wild-type (WT) mice. Moreover, PirB[−/−] mice show diminished encephalitis and improved survival following T3 reovirus infection. Collectively, these data suggest that PirB functions as a binding and internalization receptor for reovirus and promotes T3 reovirus replication and pathogenesis in the murine brain. Thus, this study identifies a bona fide neural receptor for reovirus, which broadens the spectrum of reovirus receptors and deepens an understanding of how receptor expression regulates reovirus replication and virulence.

## Results

### CRISPRa screen for reovirus receptors identifies PirB as a candidate

Compared with loss-of-function approaches for receptor identification, gain-of-function approaches do not require receptor-expressing cell lines and are insensitive to receptor redundancy. We conducted a screen for reovirus receptors using CRISPRa methodology (Fig. 1a), hypothesizing that activation of receptor expression would enhance the reovirus binding capacity of cells. To avoid potential confounding binding to known receptors, we engineered an immortalized JAM-A[−/−]/NgR1[−/−] double-knockout (DKO) mouse embryonic fibroblast (MEF) cell line, which displays diminished reovirus binding capacity (Supplementary Fig. 1) and thus serves as a suitable cell line for CRISPRa screening. To eliminate the confounder of SA-mediated binding, we used glycan-blind reovirus strains, T1SA-[34,41,42] and T3SA-[27,34,41], which are incapable of binding SA. MEFs first were engineered to express dCas9 and then transduced with lentiviruses expressing a murine genome-wide CRISPRa library. Transduced cells were screened for

reovirus binding and sorted for receptor-expressing cells. sgRNAs were identified by deep sequencing and filtered based on fold change and subcellular localization of the corresponding protein (Fig. 1b).

The most highly enriched candidate was PirB, a member of the LILR family that includes immune stimulatory receptors, LILRAs, and immune inhibitory receptors, LILRBs[35,39,43,44]. PirB is the sole murine LILRB receptor. In contrast, multiple murine LILRA orthologs, with the prototype called paired immunoglobulin-like receptor A (PirA), have been identified. LILRs are expressed by hematopoietic cells and transduce immune activating or inhibitory signaling following binding to MHC class I molecules. Human LILRB2 and mouse PirB also are expressed by neurons and function as receptors for MHC-I molecules and myelin-associated inhibitors (MAIs), including Nogo66, myelin-associated glycoprotein, and oligodendrocyte-myelin glycoprotein. Remarkably, PirB is bound by the three MAI ligands that also engage known reovirus receptor NgR1[22].

### Ectopic expression of PirB confers reovirus binding to and infection of non-susceptible cells

To validate PirB as a receptor for reovirus, we first analyzed reovirus binding following transfection of PirB cDNA into Chinese hamster ovary (CHO) cells (Fig. 2a), which do not allow reovirus binding and infection[22,34]. We also tested reovirus binding to CHO cells transfected with cDNAs encoding human PirB homolog LILRB2 and mouse paralog PirA. Known reovirus receptor JAM-A and coxsackievirus and adenovirus receptor (CAR) were used as positive and negative controls, respectively. Consistent with the screen results, reovirus strains T1SA- and T3SA- were capable of binding PirB-expressing cells (Fig. 2b and Supplementary Fig. 2a). We detected weak reovirus binding to LILRB2-expressing cells but no detectable binding to PirA-expressing cells. We next tested whether PirB-mediated reovirus binding allowed infection (Fig. 2c and Supplementary Fig. 2b). Both T1SA- and T3SA- were capable of infecting PirB-expressing cells to a level similar to that of JAM-A-expressing cells. In contrast, PirA expression resulted in few infected cells, and LILRB2 expression led to an intermediate phenotype.

To validate a function of the PirB extracellular region in reovirus attachment, we tested whether a PirB-specific antibody interferes with reovirus binding and infection. PirB-expressing cells were incubated with increasing concentrations of a PirA/B-specific mAb (6C1) or an isotype control prior to reovirus adsorption (Fig. 2d, e). While treatment with the isotype control had no demonstrable effect, mAb 6C1 treatment led to a dose-dependent decrease in reovirus binding and infectivity. Additionally, we tested the effect of prior-to-adsorption incubation of the virus with recombinant PirB ectodomain on the infectivity of PirB-expressing cells (Fig. 2f). Consistent with results of the antibody-blockade assay, recombinant PirB ectodomain diminished reovirus infectivity in a dose-dependent manner. These findings indicate that ectopically expressed PirB promotes reovirus binding and infection, suggesting that PirB is a reovirus receptor.

### PirB D3D4 domains are required for reovirus binding and infection

To define regions in the PirB ectodomain required for reovirus binding and infection, we engineered chimeric receptor proteins by reciprocally exchanging structurally homologous domains of PirA and PirB (Fig. 3a). We hypothesized that exchanging essential PirB sequences with non-essential PirA sequences would diminish reovirus binding and infection and vice versa. Ectodomains of PirA and PirB contain six immunoglobulin (Ig)-like domains (D1-D6) and share substantial amino acid identity (~92%)[35]. Surface expression of the chimeric receptors was comparable to that of WT PirA and PirB (Supplementary Fig. 3). Exchanging PirB D1D2 or D5D6 with the corresponding regions of PirA (PirB A-D1D2 and PirB A-D5D6) did not decrease reovirus binding relative to WT PirB (Fig. 3b). In contrast, exchanging PirB D1D4 or D3D4 with the corresponding regions of PirA (PirB A-D1D4 and PirB A-D3D4)

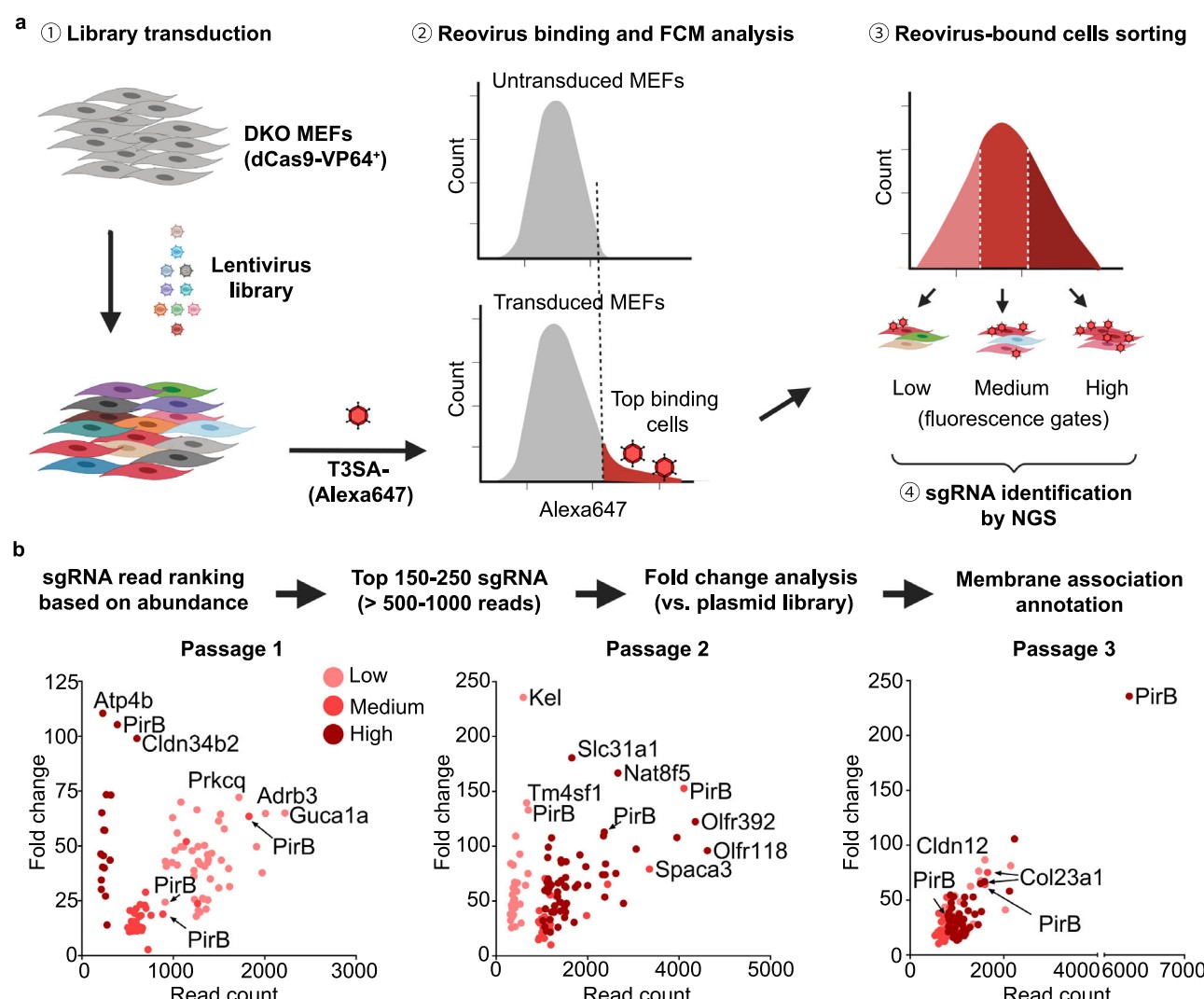

**Fig. 1 | CRISPR activation screen identifies PirB as a potential host receptor for reovirus. a** Schematic of CRISPRa screening methodology. ① JAM-A⁻/⁻ x NgR1⁻/⁻ double-knockout (DKO) MEFs stably expressing dCas9-VP64 were transduced with lentiviruses encoding a murine genome-wide CRISPRa library. Transduced MEFs were serially passaged three times. ② Binding of Alexa-647-labeled reovirus strain T3SA- to transduced DKO MEFs was assessed by flow cytometry. ③ The ~ 1% most fluorescent cells were sorted into three populations based on low, medium, and high median fluorescence intensity. ④ Genomic DNA from each cell population was analyzed by next-generation sequencing (NGS) to identify corresponding sgRNA

sequences. Experiments were conducted using sublibrary A and B. The diagram was prepared using BioRender. **b** Bioinformatic analysis of screen results. sgRNAs were ranked by read abundance and fold change compared with the input library. Candidate receptor genes encoding proteins with known plasma membrane distribution were selected for validation. Receptor candidate lists of three cell passages of sublibrary A screening are depicted in dot plots. Candidates from low, medium, and high fluorescence intensity sorting are highlighted with light, medium, and dark red colors, respectively.

diminished reovirus binding to a level comparable to WT PirA. Concordantly, replacement of PirA D1D4 or D3D4 with the corresponding regions of PirB (PirA B-D1D4 and PirA B-D3D4) enhanced reovirus binding, while replacement of PirA D1D2 with PirB D1D2 (PirA B-D1D2) did not. Interestingly, chimeras PirB A-D1D2 and PirA B-D3D4 have the same (PirB) D3D4 region and allow substantially higher levels of reovirus binding than WT PirB. Furthermore, these differences in reovirus binding correlated well with the capacity of the chimeric receptors to mediate infection (Fig. 3c). Collectively, these results indicate that exchange of D3D4 but not D1D2 or D5D6 yields a phenotypic switch between PirA and PirB, suggesting that the PirB D3D4 domains are required for reovirus binding and infection and provide additional evidence that PirB is a reovirus receptor.

**Biophysics of the reovirus and PirB interaction**

To confirm a function for PirB as a bona fide reovirus receptor, we tested whether reovirus directly interacts with PirB using atomic force

microscopy (AFM) and analyzed the kinetics and thermodynamics of the interaction[19–21]. By functionalizing tips with reovirus T3SA- virions, the binding probability was assessed by scanning gold-coated model surfaces grafted with recombinant PirB proteins (Fig. 4a). By approaching and retracting the functionalized tip to and from the surface repeatedly, force-distance curves could be obtained and used to assess the frequency of specific binding events based on the adhesion events present within the retrieved curves[45]. Substantial binding was observed for T3SA- virions, which was significantly higher than virion binding to the Ni²⁺-NTA control surface (Fig. 4b). Following incubation with a PirB-specific mAb, the binding probability decreased to a level statistically indistinct from interactions with the control surface, suggesting interaction specificity. We then assessed the kinetic properties of reovirus bond formation with PirB (Fig. 4c and Supplementary Fig. 4). By varying the retraction speed of the AFM tip and thus the mechanical force applied over time, we examined force as a function of loading rate. By dividing the retrieved data into the

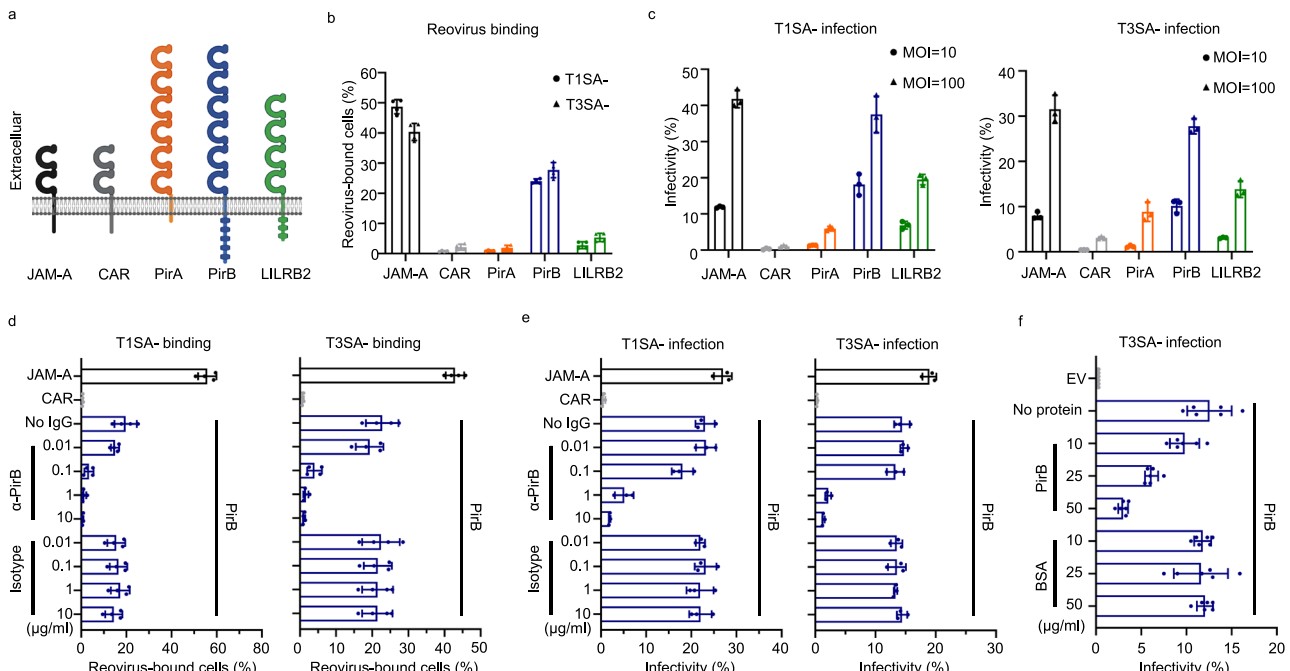

**Fig. 2 | PirB promotes reovirus binding and infection. a** Domain organization of host proteins used in assays of reovirus receptor binding. The diagram was prepared using BioRender. **b** Reovirus binding to receptor-expressing cells. CHO cells were transfected with the cDNAs shown and adsorbed with Alexa-647-labeled reovirus T1SA- or T3SA-. Virus-bound cells were quantified by flow cytometry. JAM-A and CAR were used as positive and negative controls, respectively. **c** Reovirus infection of receptor-expressing cells. Transfected cells expressing the cDNAs shown were adsorbed with T1SA- or T3SA- at a multiplicity of infection (MOI) of 10 or 100 PFU/cell. **d** Effect of PirB-specific antibody on reovirus binding. PirB-expressing cells were incubated with the concentrations shown of PirA/B-specific monoclonal antibody (mAb) 6C1 or isotype control IgG and adsorbed with

Alexa-647-labeled T1SA- or T3SA-. **e** Effect of PirB-specific antibody on reovirus infection. CHO cells were incubated with mAb 6C1 or isotype IgG and adsorbed with T1SA- or T3SA- (MOI of 50 PFU/cell). **f** Effect of recombinant PirB ectodomain incubation with virus on reovirus infection. CHO cells were transfected with PirB cDNA or empty vector (EV). T3SA- was incubated with recombinant PirB ectodomain (PirB D1D6) or bovine serum albumin (BSA) prior to adsorption to PirB-expressing CHO cells (MOI of 50 PFU/cell). In **c**, **e**, and **f**, infected cells were quantified using an indirect immunofluorescence assay (IFA). Reovirus binding (**b** and **d**) and infectivity (**c–f**) assays were conducted in quadruplicate and triplicate, respectively. Mean values are shown. Error bars indicate standard deviation (SD).

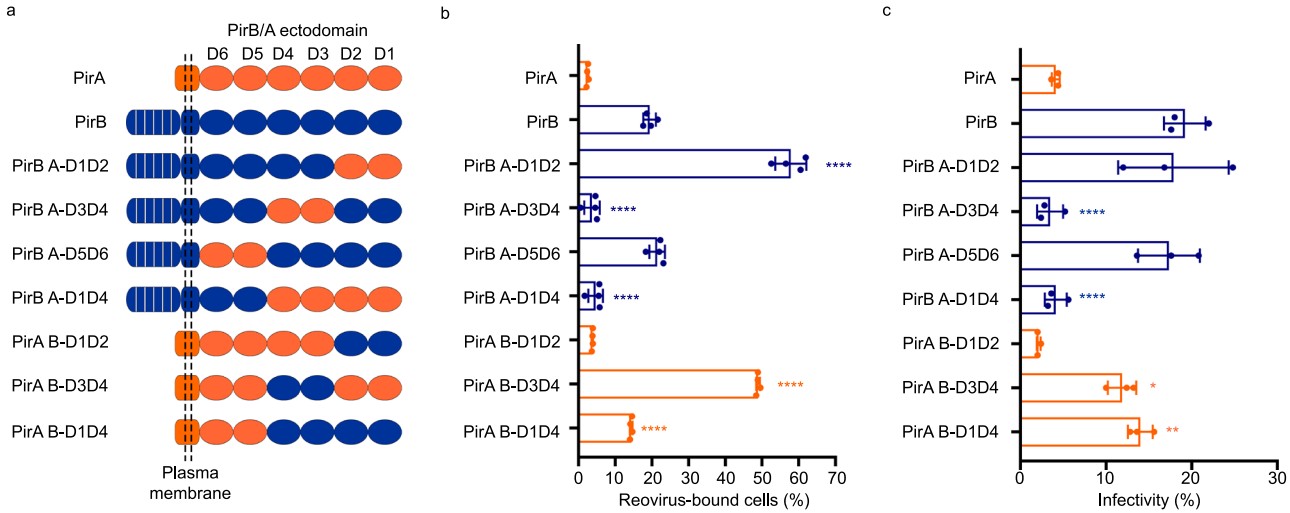

**Fig. 3 | PirB D3D4 is required for reovirus binding and infectivity. a** Schematic of reciprocal exchanges of PirA and PirB ectodomains. PirA (orange) and PirB (blue) have six homologous extracellular Ig-like domains, designated D1 to D6. PirA and PirB extracellular domain sequences were exchanged to yield chimeric receptor constructs. **b** CHO cells were transfected with the cDNAs shown and scored for reovirus T3SA- binding by flow cytometry. **c** CHO cells were transfected with the cDNAs shown, absorbed with T3SA- at an MOI of 50 PFU/cell, and scored for

infectivity by IFA. Reovirus binding (**b**) and infectivity (**c**) assays were conducted in quadruplicate and triplicate, respectively. Mean values are shown. Error bars indicate SD. Statistical analysis was conducted by comparing results of each chimeric receptor with the corresponding parental backbone, which is defined based on the transmembrane and intracellular region. *P* values were calculated using one-way ANOVA with Turkey's test. *$P < 0.05$; **$P < 0.01$; ****$P < 0.0001$.

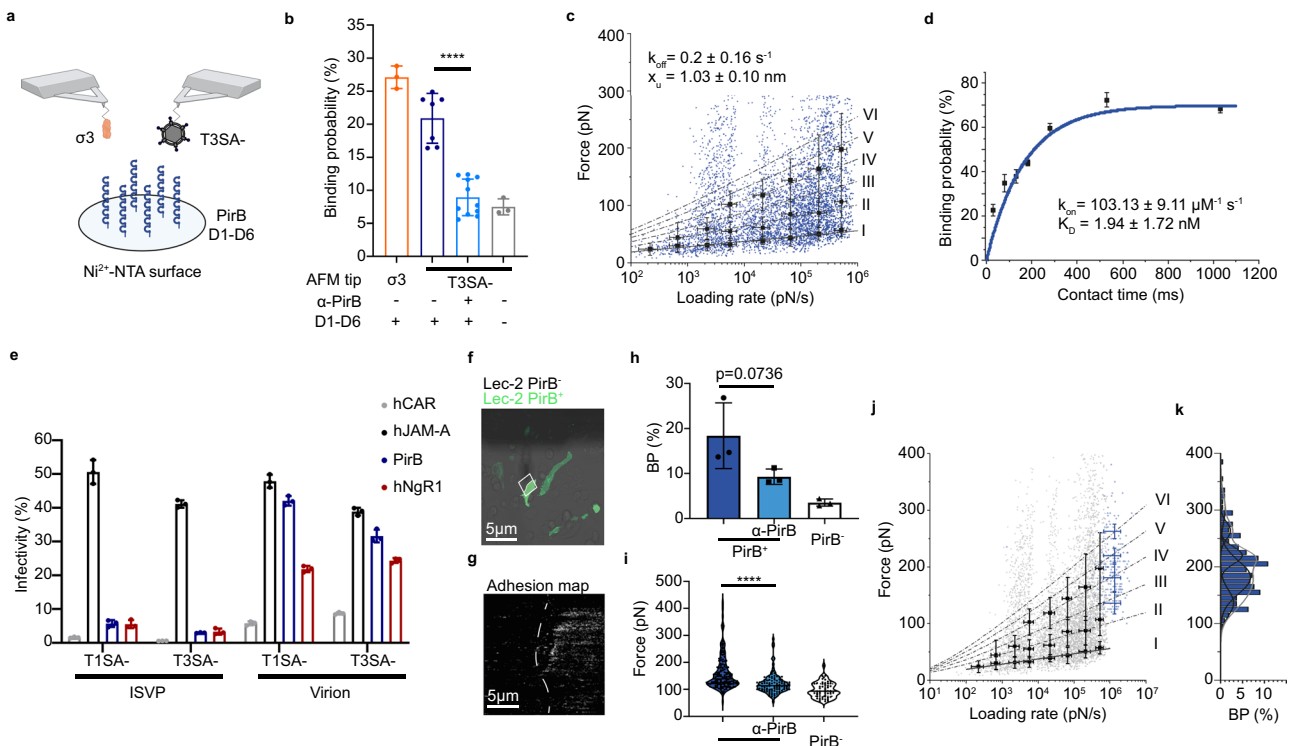

**Fig. 4 | Biophysics of reovirus-PirB interactions.** Characterization of reovirus-PirB binding thermodynamics on model surfaces (**a**–**d**) and living cells (**e**–**k**). **a** Schematic of quantifying reovirus-PirB interactions using a recombinant PirB-coated model surface. Tips were functionalized with T3SA- virions or recombinant σ3. The diagram was prepared using BioRender. **b** Binding probability on model surfaces with or without PirB-specific mAb treatment. Uncoated surface, non-specific control. σ3-PirB, $N = 3$; virion-PirB, $N = 7$; virion-PirB plus mAb, $N = 11$; uncoating surface, $N = 3$. **c** Dynamic force spectroscopy (DFS) plot of the distribution of average rupture forces across eight discrete loading rate ranges. Data corresponding to single and multivalent interactions fit Bell-Evans (solid line) and Williams-Evans models (dotted lines). $N = 6605$. **d** Binding probability based on the contact time of reovirus-functionalized tips with a PirB-coated surface. Least-squares fit of the data to a mono-exponential decay model (blue line, $r^2$ of 0.99)

provides the average binding kinetic on-rate ($k_{on}$). $N = 5$. **e** Susceptibility of PirB-expressing cells to infection by reovirus virions and ISVPs. Virion, 100 PFU/cell; ISVP, 10,000 ISVPs/cell. $N = 3$. **f** Confocal micrograph of PirB-2A-GFP-expressing Lec2 cells. **g** Representative adhesion map from the boxed area in (**f**). Reovirus binding is indicated by gray-to-white pixels. In **f** and **g**, scale bar, 5 µm. **h** Reovirus-PirB binding probability using living cells. $N = 3$. **i** Adhesion force of reovirus binding to live cells. Virion-PirB, $N = 182$; virion-PirB plus mAb, $N = 140$; uncoating surface, $N = 93$. **j** DFS plot of prior model surface data (gray) incorporating live-cell data (blue). $N = 6605$. **k** Histogram of the force distribution of the live-cell data and a multi-peak Gaussian fit. $N = 225$. In **b**–**e**, **h**–**j**, mean values are shown. In **b**–**e**, **h**–**j**, error bars indicate SD. In **b**, **h**, and **i**, $P$ values were calculated using one-way ANOVA with Turkey's test. ****$P < 0.0001$.

loading rate ranges, we fit the data to the Bell-Evans model to discern both the dissociation rate ($k_{off}$) and distance to the transition state ($x_u$) for this interaction[46–48]. For the reovirus-PirB complex, we obtained a $k_{off}$ of $0.20 \pm 0.16 \, s^{-1}$ (mean ± standard deviation [SD]) and an $x_u$ of $1.03 \pm 0.10 \, nm$ (mean ± SD), suggesting that although there is a relatively large degree of conformational flexibility for this interaction, once established, the supramolecular bonds are stable. Higher forces in the dynamic force spectroscopy plot correlate well with Williams-Evans predictions for multiple uncorrelated bonds[49], indicating that reovirus forms multivalent bonds with PirB. By varying the time during which the tip was brought into contact with the PirB model surface and monitoring changes in binding probability, and working under the assumption that the reovirus-PirB complex can be approximated using pseudo-first-order kinetics, we were able to estimate the association rate ($k_{on}$)[50]. As shown in Fig. 4d, the data fit well with this model, providing a $k_{on}$ of $103.13 \pm 9.11 \, µM^{-1} \, s^{-1}$ (mean ± SD). This information was used to calculate an equilibrium dissociation constant ($K_D$) for the reovirus-PirB interaction of $1.94 \pm 1.72 \, nM$ (mean ± SD), which suggests that the interaction between reovirus and PirB is specific and approximates the $K_D$ values of other reovirus entry receptors characterized previously[19,21,23].

Since PirB engages the same MAIs as NgR1, we tested whether PirB binds the reovirus σ3 outer-capsid protein, which also is bound by NgR1[19]. We detected a high binding probability of recombinant σ3

protein with PirB ($27.1 \pm 1.7\%$) (mean ± SD) (Fig. 4b), suggesting that reovirus σ3 is responsible for supramolecular bond formation between reovirus and PirB. We then tested whether reovirus infectious sub-virion particles (ISVPs), which lack σ3[17], are capable of infecting PirB-expressing cells (Fig. 4e). Both virions and ISVPs can infect cells expressing JAM-A, but only virions can infect cells expressing NgR1[22,24]. Similar to NgR1-expressing cells, reovirus virions but not ISVPs can infect PirB-expressing cells. These results suggest that outer-capsid protein σ3 is the reovirus ligand for PirB.

To validate that the reovirus-PirB interaction on a model surface occurs in a cellular context, we analyzed reovirus binding to living Lec2 cells by AFM (Fig. 4f). Lec2 cells express significantly less cell-surface SA and do not express other known reovirus receptors[21]. Lec2 cells transiently transfected with PirB were probed by T3SA- functionalized tips. Images were acquired of PirB-expressing cells immediately adjacent to non-transfected cells for internal controls in each adhesion map (Fig. 4g). Using pixel counting, binding probabilities (Fig. 4h) and adhesion force (Fig. 4i) were calculated for each population and yielded ample reovirus binding to PirB-transfected cells, which is comparable to the binding to PirB model surfaces. To compare the binding kinetics on living cells with model surfaces, we extracted the force and loading rate from the force *versus* time curves obtained and overlaid these data with results obtained for model surfaces and plotted the retrieved forces as histograms (Fig. 4j and Fig. 4k, respectively). The

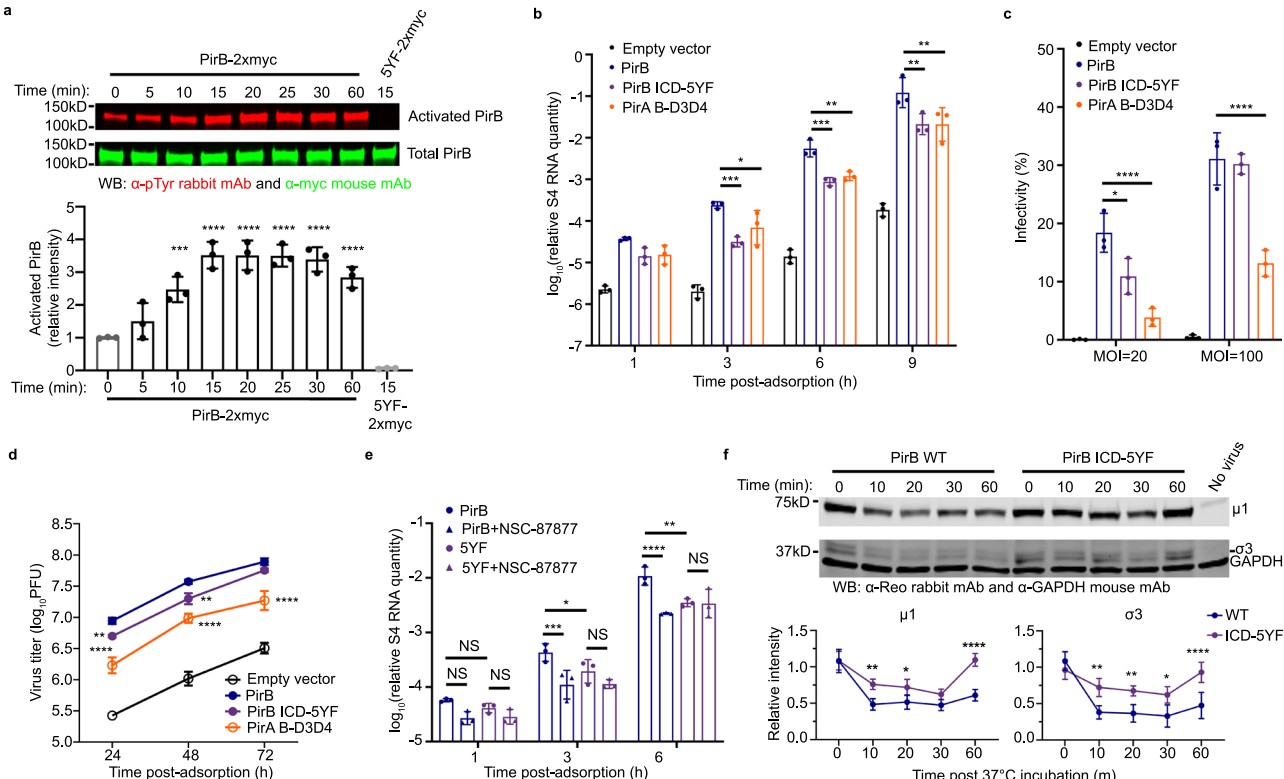

**Fig. 5 | PirB signaling is required for efficient reovirus entry. a–f** CHO cells were transfected with the cDNAs shown. **a** PirB signaling induced by reovirus binding. Cells were treated with $Na_3VO_4$ prior to and during T3SA+ adsorption. PirB was immunoprecipitated and detected by immunoblotting. Relative pTyr intensity was determined by normalizing to total PirB signal intensity and comparing with the intensity of the pre-adsorption signal (0 min). **b** Reovirus transcription following infection of WT PirB- or mutant-expressing cells. **c** Reovirus infection of WT PirB or mutant-expressing cells. Cells were adsorbed with T3SA- (20 or 100 PFU/cell). Infectivity was quantified by IFA at 24 hpa. **d** Reovirus infection of WT PirB or mutant-expressing cells. Cells were adsorbed with T3SA- (20 PFU/cell). Viral titers in cell lysates were determined by plaque assay. **e** Effect of SHP-1/2 inhibitor on reovirus internalization into WT or mutant PirB-expressing cells. Cells were incubated with NSC-87877 prior to and during T3SA- adsorption. In **b** and **e**, cells were adsorbed with T3SA- and analyzed for s4 RNA levels in cell lysates at the times shown by RT-qPCR. Levels of s4 RNA were normalized to levels of β-actin RNA. **f** Reovirus uncoating in cells expressing WT or mutant PirB. CHO cells were adsorbed with T3SA+ on ice and incubated at 37 °C for the times shown. Viral proteins were detected by immunoblotting. Levels of µ1 and σ3 proteins were normalized to levels of glyceraldehyde 3-phosphate dehydrogenase (GAPDH) and compared with the corresponding levels pre-uncoating (0 min). In **a–e** and **f**, experiments were conducted in triplicate and quadruplicate, respectively. Mean values are shown. Error bars indicate SD. $P$ values were calculated using one-way ANOVA with Turkey's test (**a**) and two-way ANOVA with Holm-Sidak's test (**b–f**). *$P < 0.05$; **$P < 0.01$; ***$P < 0.001$; ****$P < 0.0001$.

information gathered from the live-cell experiments aligned well with data from the model surfaces, supporting the relevance of the model surface data to a biological context. Collectively, biophysical characterization indicates that the reovirus-PirB interaction has high affinity and specificity and suggests that reovirus σ3 interacts with PirB.

## PirB intracellular signaling is required for efficient reovirus entry

LILRBs including PirB signal through the activation of multiple intracellular domain (ICD) motifs[35,39,44]. The PirB ICD contains four immunoreceptor tyrosine-based inhibitory motifs (ITIMs) and one potential immunoreceptor tyrosine-based switch motif (ITSM)[51]. To determine whether PirB signaling is required for reovirus entry, we constructed a PirB signaling-deficient mutant (ICD-5YF), in which the five tyrosine residues within the ICD capable of potential phosphorylation (pTyr) were exchanged with phenylalanine residues (Fig. 5 and Supplementary Fig. 5). We also used a PirA-PirB chimera (PirA B-D3D4) (Fig. 5b–d and Supplementary Fig. 5) that is capable of binding reovirus (Fig. 3) but has a PirA cytoplasmic tail. WT PirB and the ICD mutant receptors promote comparable reovirus binding (Supplementary Fig. 5b), suggesting that the PirB ICD does not function in reovirus attachment. We first tested whether reovirus binding activates PirB signaling in CHO cells transfected with either WT PirB or the ICD-5YF mutant. PirB was captured by immunoprecipitation and probed for phosphorylation

using a pTyr-specific mAb cocktail (Fig. 5a). In this assay, glycan-binding reovirus (T3SA+) was used to enhance the interaction with PirB to facilitate detection of transient phosphorylation. Levels of pTyr-PirB increased more than three-fold following reovirus adsorption relative to unbound PirB. As anticipated, there was no detectable phosphorylation of the ICD-5YF mutant. These findings indicate that PirB signaling is activated following reovirus binding.

To investigate the importance of PirB signaling in reovirus entry, we quantified levels of S4 gene RNA by RT-qPCR following viral adsorption to CHO cells expressing WT PirB or the ICD mutant receptors (Fig. 5b). Viral RNAs detected at 1 h post-adsorption (hpa) are primarily genomic RNAs contained in internalized particles[52]. As expected, we did not detect significant differences in S4 RNA levels in cells transfected with WT PirB and the ICD mutants at 1 hpa. However, at 3, 6, and 9 hpa, S4 RNA levels in cells expressing WT PirB were significantly higher than those in cells expressing PirB ICD-5YF or PirA B-D3D4, suggesting that entry steps leading to viral transcription require a functional PirB cytoplasmic domain. Concordantly, reovirus infectivity of WT PirB-expressing cells was substantially greater than those expressing ICD mutants (Fig. 5c, d). pTyr resides within ITIMs provide docking sites for signal-transducing phosphatases SHP-1/2[39,43]. Treatment of cells expressing WT PirB with SHP-1/2 inhibitor NSC-87877[53] also decreased viral transcription. However, NSC-87877 treatment of PirB ICD-5YF mutant-expressing cells did not alter viral

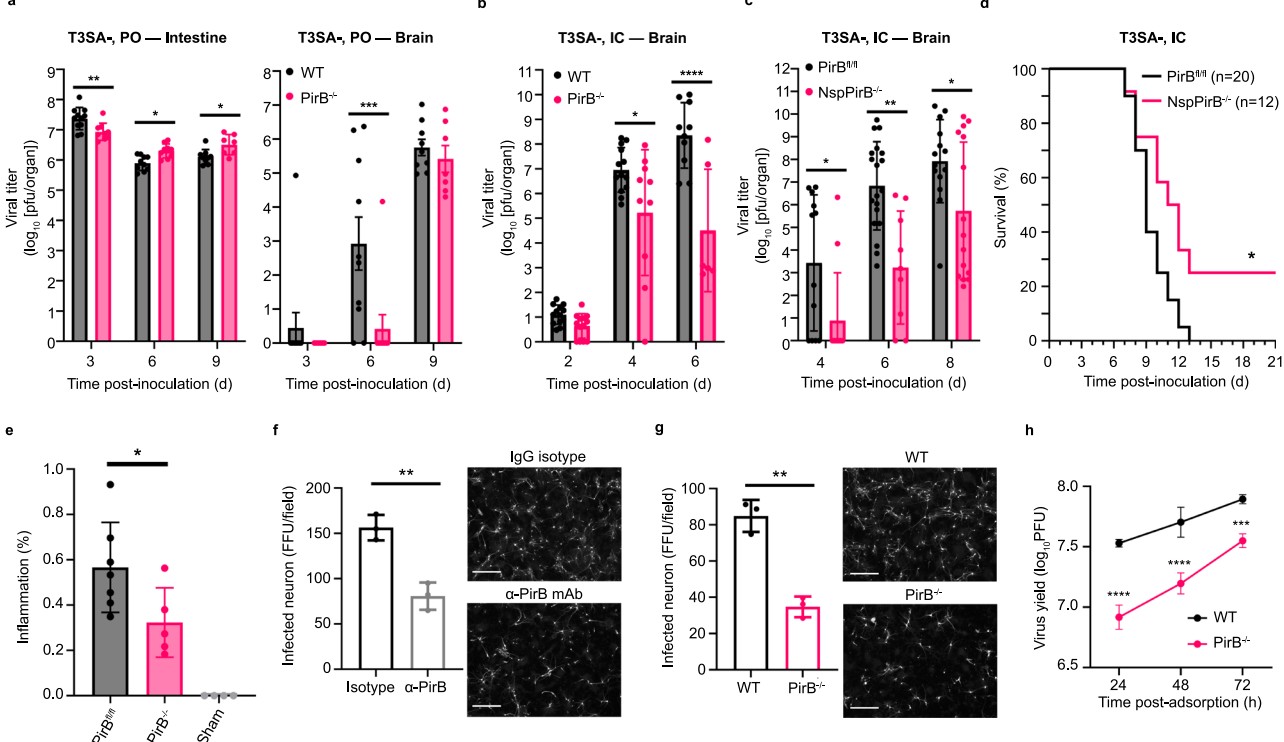

**Fig. 6 | PirB is required for efficient T3 reovirus replication and pathogenicity in the murine CNS. a** T3 reovirus replication in WT and PirB$^{-/-}$ mice. Mice were inoculated perorally with T3SA- ($10^4$ PFU/mouse). $N = 11/10/9$ at 3/6/9 DPI (WT); $N = 9/9/7$ at 3/6/9 DPI (PirB$^{-/-}$). **b** T3 reovirus replication in the brain of WT and PirB$^{-/-}$ mice. $N = 13/13/10$ at 2/4/6 DPI (WT); $N = 13/10/6$ at 2/4/6 DPI (PirB$^{-/-}$). **c** T3 reovirus replication in the brain of PirB$^{fl/fl}$ and NspPirB$^{-/-}$ mice. $N = 13/19/14$ at 2/4/6 DPI (PirB$^{fl/fl}$); $N = 12/9/15$ at 2/4/6 DPI (NspPirB$^{-/-}$). **d** T3 reovirus virulence in PirB$^{fl/fl}$ and NspPirB$^{-/-}$ mice. **e** Encephalitis following infection of PirB$^{fl/fl}$ and NspPirB$^{-/-}$ mice. PBS-inoculated PirB$^{fl/fl}$ mice, sham control. Brain inflammation was monitored by MRI at 8 dpi and defined as hyperintensity. Relative level is indicated by hyperintensity voxel percentage. $N = 7$ (PirB$^{fl/fl}$); $N = 5$ (NspPirB$^{-/-}$); $N = 4$ (sham). In **b**–**e**, mice were inoculated intracranially with T3SA- (25 PFU/mouse). **f** Reovirus

infection of primary cortical neurons is blocked by PirB-specific mAb. Neurons were pre-incubated with mAb or isotype IgG and adsorbed with T3SA+ (100 PFU/cell). **g, h** T3 reovirus infection of WT and PirB$^{-/-}$ primary neurons. Primary murine cortical neurons were adsorbed with T3SA+ (20 PFU/cell) (**g**). In **f** and **g**, scale bar, 150 μm. Viral titers in tissue (**a**–**c** and **e**) and neuron lysates (**h**) were determined by plaque assay. In **f**–**h**, experiments were conducted in triplicate. WT and PirB$^{-/-}$ mice have comparable tissue weights. In **a**–**c** and **e**, each symbol indicates a single mouse. Mean values are shown. Error bars indicate SD. Statistical analysis: **a**–**c** and **h**, two-way ANOVA with Holm-Sidak's test; **d** log-rank (Mantel-Cox) test; **e** one-way ANOVA with Turkey's test; **f** and **g** two-sided Student's t-test. *$P < 0.05$; **$P < 0.01$; ***$P < 0.001$; ****$P < 0.0001$.

transcript levels (Fig. 5e). Collectively, these data suggest that PirB signaling, possibly transduced by SHP-1/2 phosphatases, functions in reovirus entry.

To further evaluate a role for PirB signaling in post-attachment steps, we compared reovirus uncoating kinetics in cells expressing WT or ICD-5YF PirB (Fig. 5f). Virion-to-ISVP disassembly in the endocytic pathway is characterized by the cleavage of outer-capsid protein μ1 and the complete loss of σ3[17]. Reovirus uncoating occurred efficiently during entry into cells expressing WT PirB, but disruption of PirB signaling significantly attenuated reovirus uncoating (Fig. 5f). These findings are concordant with the results of viral RNA transcription (Fig. 5b) and infectivity (Fig. 5c, d) experiments. Collectively, these data suggest that PirB signaling activated by reovirus attachment is required for efficient reovirus entry.

## PirB contributes to T3 reovirus replication and pathogenicity in the murine CNS

To determine whether PirB is required for reovirus replication and pathogenesis in vivo, we compared the susceptibility of WT and PirB-null (PirB$^{-/-}$) mice to reovirus infection and disease. Peroral (PO) inoculation mimics the natural fecal-oral transmission route, in which reovirus establishes primary replication in the intestine and spreads to sites of secondary replication, including the brain[17,18]. Mice were inoculated perorally with T1SA- or T3SA-, and viral replication in the

intestine and sites of secondary replication including the brain were quantified (Supplementary Fig. 6 and Fig. 6a). Titers of T1SA- were comparable in WT and PirB$^{-/-}$ mice at 3, 6, and 9 days post-inoculation (dpi) in all tissues tested (Supplementary Fig. 6a), suggesting that PirB is not required for replication of T1 reovirus in mice. Titers of T3SA- were comparable in the intestine of WT and PirB$^{-/-}$ mice at 3, 6, and 9 dpi. However, titers in the brain of WT mice were significantly higher than those in PirB$^{-/-}$ mice at 6 dpi (Fig. 6a), at which viral titer was detected in a single PirB$^{-/-}$ pup of 10 tested. These data suggest that dissemination of T3SA- from the intestine to the brain of PirB$^{-/-}$ mice, T3SA- replication at that site, or both are dependent on PirB. We observed a similar trend toward decreased T3SA- titers in the heart, lung, liver, and spleen of PirB$^{-/-}$ mice relative to WT mice at 6 dpi, with the greatest difference observed in viral titers in the heart (Supplementary Fig. 6b). To test whether PirB is required for replication of T3 reovirus in the brain, we inoculated newborn mice intracranially with T3SA- (Fig. 6b). Intracranial (IC) inoculation circumvents primary replication in the intestine and dissemination routes to the brain. Titers of T3SA- in the brain of WT mice were significantly higher than those in PirB$^{-/-}$ mice at 4 and 6 dpi. Collectively, these results suggest that PirB is required for efficient T3 reovirus replication in the murine brain.

PirB is expressed by CNS neurons[38,40] as well as hematopoietic cells that may traffic to the brain[35,39,43,44]. To determine whether PirB is

neural receptor for T3 reovirus, we established neural-specific PirB-null mice (NspPirB$^{-/-}$). Mice with PirB alleles flanked by loxP sequences (PirB$^{fl/fl}$)[38,54] were interbred with mice expressing Cre recombinase under control of a nestin (neural-specific) promotor to obtain NspPirB$^{-/-}$ mice[55]. Newborn PirB$^{fl/fl}$ and NspPirB$^{-/-}$ mice were inoculated intracranially with T3SA- reovirus, and viral loads in the brain were determined at 4, 6, and 8 dpi. Titers of T3SA- in the brain of PirB$^{fl/fl}$ mice were significantly higher than those in NspPirB$^{-/-}$ mice at all three timepoints, with the greatest difference at 6 dpi (Fig. 6c), concordant with results following IC inoculation of WT and PirB$^{-/-}$ mice (Fig. 6b). To test whether neural-specific PirB is required for reovirus neurovirulence, PirB$^{fl/fl}$ and NspPirB$^{-/-}$ mice were inoculated intracranially with T3SA- and monitored for survival for 21 d (Fig. 6d). In this experiment, 25% of NspPirB$^{-/-}$ mice survived T3SA- infection, whereas none of the PirB$^{fl/fl}$ mice survived, indicating that PirB expression in neural cells enhances reovirus neurovirulence. To gauge the level of brain inflammation in reovirus-infected PirB$^{fl/fl}$ and NspPirB$^{-/-}$ mice, we used magnetic resonance imaging (MRI) to quantify inflammation in the brain of living mice (Fig. 6e and Supplementary Fig. 7). Relative to the brain of PirB$^{fl/fl}$ mice, significantly less reovirus-induced inflammation was apparent in the brain of NspPirB$^{-/-}$ mice, in agreement with the improved survival (Fig. 6d). To test a potential role for PirB in reovirus tropism in the CNS, we compared reovirus antigen distribution in brain sections of intracranially inoculated PirB$^{fl/fl}$ and NspPirB$^{-/-}$ mice (Supplementary Fig. 8). The staining results show that reovirus displays similar tropism in the brain of PirB$^{fl/fl}$ and NspPirB$^{-/-}$ mice, in which viral proteins are detected in the cortex, hippocampus CA2/3 region, thalamus, hypothalamus, pons, and cerebellum[33,34,41]. Therefore, PirB does not appear to contribute to reovirus tropism in the murine brain.

In a final series of experiments, we tested whether reovirus infection of primary cultures of cortical neurons requires expression of PirB. Primary cortical neurons were cultivated from WT mice, incubated with a PirB-specific or control antibody, adsorbed with T3 reovirus, and scored for infectivity (Fig. 6f). PirB-specific antibody decreased T3 reovirus infection of primary neurons by ~50%. We also compared the capacity of T3 reovirus to infect neurons cultivated from WT and PirB$^{-/-}$ mice (Fig. 6g, h). Concordantly, T3 reovirus replication in PirB$^{-/-}$ neurons was significantly attenuated relative to that of WT neurons. These data correlate well with the diminished T3 reovirus replication in the brain of PirB$^{-/-}$ mice (Fig. 6a–c) and suggest that PirB is a neuronal receptor for T3 reovirus in the murine brain.

## Discussion

Reovirus proteins that form the outer capsid include 12 σ1 trimers embedded in 12 λ2 pentamers at the icosahedral five-fold axes and 200 σ3-μ1 heterohexamers that form the bulk of the capsid[17]. Reovirus serotype-dependent CNS tropism is determined by the head domain of σ1 protein[41]. We identified PirB as a reovirus neural receptor using a gain-of-function CRISPRa screen. Reovirus σ3 is the viral ligand for PirB, similar to the binding mechanism for NgR1[19]. The identification of PirB as a reovirus receptor further supports the concept that reovirus entry is coordinated by multiple host factors.

PirB functions as a receptor for both T1 and T3 reovirus strains and, therefore, it is not the primary neural tropism determinant. However, PirB is required for efficient T3 reovirus replication and pathogenicity in the murine CNS. Relative to the σ1 protein, the high copy number of virion-associated σ3 protein would allow more numerous multivalent interactions with host receptors and could facilitate reovirus attachment and entry. Moreover, not all orthoreoviruses encode a σ1-like adhesion fiber, but all encode a capsid component analogous to σ3[56]. We speculate that host receptors engaged by σ3 homologs of orthoreoviruses lacking adhesion fibers may serve a more significant role in viral entry and pathogenesis. Results presented here identify PirB as a reovirus receptor on neurons

and help elucidate mechanisms governing reovirus neurotropism and neuropathogenesis.

The multiple viral capsid components and host factors that contribute to a step-wise entry pathway for reovirus also may contribute to its broad host range and tissue tropism. The initial contact between reovirus and host cells is likely to occur by low-affinity binding of σ1 to SA[21,27,42], which adheres the virus to the cell surface and induces conformational changes in σ1. The conformational changes in σ1 allow the head domain to engage receptors with higher affinity[23,57]. The exposed σ1 head domain binds JAM-A[23], which is expressed by epithelial and endothelial cells as well as by leukocytes[33], and possibly serotype-specific receptors expressed on ependymal cells (for T1 reovirus) and neurons (for T3 reovirus)[18,41]. Outer-capsid protein σ3 binds additional receptors, including NgR1 in human neurons[22,34] and PirB in mouse neurons (Fig. 6f–h). The high avidity of clustered σ3-NgR1 or σ3-PirB interactions appears to activate receptor signaling to promote virus endocytosis (Figs. 4 and 5). Interactions between λ2 and β1 integrins also facilitate reovirus internalization by recruiting clathrin for endocytosis, at least in some cell types[20,29,30]. CHO cells used for ectopic expression of PirB do not express σ1 receptors. Reovirus efficiently binds and infects PirB-expressing CHO cells, suggesting that σ1-receptor interactions are not required for interactions with PirB. In unpublished work, we compared internalization into PirB-expressing cells of WT virus and a mutant with disrupted λ2 integrin-binding motifs. There was no significant decrease in entry efficiency of the mutant virus, suggesting that λ2-integrin interactions also are not required for σ3-PirB interactions. In this model of reovirus attachment and internalization, the unidentified serotype-dependent σ1 receptors dictate reovirus tropism in the CNS. Our findings suggest that neuronal PirB is required for maximal reovirus replication in the brain by promoting viral attachment and internalization, perhaps following initial engagement of σ1 receptors.

We have identified two structurally distinct σ3 receptors, human NgR1 and murine PirB, that bind the same natural ligands, MAIs[40]. NgR1 is a GPI-linked protein and binds MAIs as part of a receptor complex[58]. It is not known whether PirB participates in a receptor complex, but it interacts with one key component of the NgR1 complex, the p75 neurotrophin receptor[59]. Considering the broad host range of mammalian orthoreovirus, mammalian MAI receptors NgR1 and PirB may function as cross-species receptors for reovirus neural infection in mammals. Furthermore, the differential preference for host receptors by reovirus suggests species-specific receptor selection by reovirus infection. Hence, evolutionary analysis of σ3 receptor use in diverse mammalian hosts may provide clues about the genetic determinants of host susceptibility to reovirus infection and potential roles of reovirus infection in mammalian evolution. MAI receptors expressed by immature neurons of the developing brain are not bound to native ligands, but they are bound to these ligands as neurons mature[60]. Therefore, it is possible that MAI receptors contribute to the strict age-restriction of lethal encephalitis caused by reovirus in young mammals[17,18].

Infection experiments using germline PirB$^{-/-}$ and NspPirB$^{-/-}$ mice indicate that neural PirB is required for maximal T3 reovirus replication and full neurovirulence (Fig. 6b–e). Following PO inoculation, reovirus disseminates to the CNS using both hematogenous and neural routes. IC inoculation circumvents the requirement for hematogenous dissemination. Viral loads in the brain are influenced by both the susceptibility of neurons to infection and the route of dissemination. Since PirB is expressed in both the CNS and peripheral nervous system (PNS), we hypothesize that PirB contributes to neural dissemination of reovirus from peripheral sites of infection to the CNS. In PirB$^{-/-}$ mice, reovirus hematogenous dissemination may not be diminished, as suggested by the comparable replication of non-neurotropic T1 reovirus in WT and PirB$^{-/-}$ mice in all tissues tested (Supplementary Fig. 6a). Therefore, at earlier times after inoculation (3 and 6 dpi),

reovirus titers are lower in the brain and other tissues of PirB[−/−] mice relative to those in WT mice (Fig. 6a and Supplementary Fig. 6b), perhaps reflecting a requirement for PirB in reovirus neural dissemination. However, at later times after inoculation (9 dpi), we think that virus transmitted using hematogenous pathways overcomes the diminished neural dissemination in PirB[−/−] mice. Therefore, our studies of the function of PirB in reovirus dissemination and tropism establish an experimental system to investigate mechanisms of enteric neurotropic virus dissemination and CNS invasion. PirB also is expressed by myeloid cells, including B cells, dendritic cells, granulocytes, macrophages, mast cells, and monocytes[61]. It is not clear whether reovirus is capable of infecting these cell types, but infectious reovirus is detectable in CD45[+] immune cells isolated from mice infected with T1 reovirus[62]. Therefore, it would be informative to investigate whether PirB serves as an entry receptor for T1 reovirus on immune cells and contributes to blockade of immunological tolerance to newly introduced food antigen, which is associated with T1 reovirus infection[63].

As observed with natural ligands[39,43,44], reovirus-PirB binding induces phosphorylation of tyrosine-based signaling motifs (Fig. 5a). LILRB receptor ITIM signaling antagonizes immune-activating ITAM signaling, especially that mediated by paired receptor LILRAs[39,44]. Several diverse pathogens bind to PirB or LILRB homologs, and this engagement of immune inhibitory receptors during viral entry is thought to attenuate antiviral innate immune responses at the initial stages of viral replication[64]. This model is exemplified by interactions between dengue virus (DENV) and LILRB1 during antibody-dependent DENV entry[65]. The expression of interferon-stimulated genes induced by immune-activation signaling of Fc receptors is dampened by DENV binding to LILRB1. Therefore, it is possible that the immune inhibitory signaling induced by reovirus binding to PirB similarly suppresses host innate immune responses elicited by reovirus entry[66] and contributes to establishment of infection. This hypothesis is supported by the exclusive preference of reovirus for PirB and not the paired immune activating receptor PirA (Fig. 2), which shares ~92% amino acid identity with PirB in the extracellular region[35].

Our discovery that reovirus attachment activates PirB intracellular signaling and facilitates reovirus endocytosis establishes that LILRB receptors function in ligand endocytosis. In neurons, PirB and LILRB2 signaling activates cofilin to depolymerize actin filaments (F-actin)[67]. Actin dynamics and mobilization are essential for ligand endocytosis[68–70], which may be a common mechanism used by PirB and LILRBs to trigger endocytosis. PirB and LILRB2 also bind other soluble ligands, including angiopoietin-related proteins[71] and amyloid-β[67]. Uptake and intracellular accumulation of amyloid-β is correlated with development of Alzheimer disease[72]. The neuronal endocytosis of reovirus and other physiological and pathological ligands may share common mechanistic features, extending the value of inquiries of PirB-mediated reovirus endocytosis. Thus, our functional characterization of the PirB extracellular and intracellular domains in reovirus entry demonstrates how multiple biological aspects of a host receptor are used by a pathogen to effect efficient entry.

Reovirus has a broad zoonotic host range[17], infecting most mammals including humans, a high prevalence in susceptible hosts[73], cross-species transmission capability[74,75], circulation in spillover-prone bat and rodent reservoirs[74,76–80], capacity for genetic reassortment[81,82], and engagement of receptors conserved across species[24,26,29]. Combined with sporadic human cases of reovirus-associated disease[74,75,80,83–85], these characteristics raise the possibility of emergence of more virulent strains, which may overcome the age restriction in disease development. Therefore, ongoing studies of reovirus pathogenesis, particularly investigation of receptor use in different host contexts, are essential for assessment of reovirus epidemic potential and countermeasure development. Structure-guided interspecies receptor comparisons will help define genetic determinants of host susceptibility, disease severity, and cross-species transmission.

Viral infection often exerts selective pressure on host receptors, especially on virus-binding surfaces[86]. Evolutionary receptor analysis for reovirus, a generalist mammalian pathogen, will determine whether reovirus infection has selected variant mammalian receptor proteins. In addition, since reovirus displays oncolytic potential and has been used in clinical trials as a cancer therapeutic[87,88], new knowledge about reovirus receptor use may foster design of more selective oncolytic agents for cancer therapy.

## Methods

### Ethics statement
All experiments in this study comply with guidelines of the U.S. Public Health Service and were approved by the Institutional Biosafety Committee at the University of Pittsburgh. All animal husbandry and experimental procedures were conducted in accordance with U.S. Public Health Service policy and approved by the Institutional Animal Care and Use Committee at the University of Pittsburgh.

### Cells and viruses
Chinese hamster ovary (CHO) cells (originally obtained from Dr. Sean Whelan, ATCC, #CCL-61) and mouse embryonic fibroblasts (MEFs) were propagated at 37 °C in 5% $CO_2$ in complete Ham's F-12 medium (GIBCO, #11765054) and Dulbecco's modified Eagle medium (DMEM) (GIBCO, #11965118), respectively. F-12 and DMEM were supplemented to contain 10% FBS (VWR, #97068-088), 2 mM L-glutamine (GIBCO, #A2916801), 100 U/ml penicillin and streptomycin (GIBCO, #10378016), and 250 ng/ml amphotericin B (Sigma-Aldrich, # A2942). Spinner-adapted L929 cells (originally obtained from Dr. Bernard Fields; ATCC CCL-1) were propagated in Joklik's modified Eagle's minimal essential medium (JMEM, United States Biological) supplemented to contain 5% FBS, 2 mM L-glutamine, 100 U/ml penicillin and streptomycin, and 250 ng/ml amphotericin B in suspension (35 °C) or monolayer (37 °C) cultures. Recombinant reoviruses, including two glycan-blind strains (T1SA- and T3SA-) and glycan-binding strain (T3SA+) were recovered by plasmid rescue using reverse genetics in previous studies[41,89]. Reovirus propagation, plaque assay titration, and ISVP preparation were conducted as described[34,41,62]. Purification of reovirus virions was conducted as described previously (dx.doi.org/10.17504/protocols.io.vj7e4rn).

### Murine embryonic fibroblast immortalization
MEFs were immortalized as described[90]. C57BL/6J mouse fetuses (E15.5) were resected from pregnant dams, heads and internal organs were removed, and the remaining tissue was sectioned into fine pieces and dissociated with 0.25% trypsin-EDTA (GIBCO, #25200114). Dissociated primary embryonic fibroblasts were propagated in tissue-culture flasks. MEFs from passage 2 were transfected with plasmid encoding SV40 large T antigen using TransIT-LT1 (Mirus Bio, #2305). Transfected MEFs were subjected to five additional passages to remove non-immortalized primary cells.

### CRISPRa library amplification and lentivirus packaging
A genome-wide mouse CRISPRa plasmid library (Caprano P65-HSF)[91] was amplified following electroporation into ElectroMAX Stbl4 competent cells (Invitrogen, #11635018) using a MicroPulser Electroporator (Bio-Rad, #1652100). Endotoxin-free library plasmids and lentivirus packaging plasmids were purified using a NucleoBond Xtra Midi EF kit (Clontech, # 740422). Lentiviruses encoding the CRISPRa library were packaged using Lenti-X 293T cells (Clontech, #632180) as described[91,92]. A total of 40 μg of library plasmid, 50 μg of second-generation psPAX2 lentiviral packaging plasmid, and 5 μg of pCMV-VSV-G envelope-expressing plasmid were transfected into a T175 tissue-culture flask (Greiner Bio-one, #660160) using TransIT-LT1 reagent. Cell supernatants were harvested at 48 and 72 h post-transfection (hpt) and incubated with concentration solution (10%

PEG 8000 and 300 mM NaCl in PBS) overnight to precipitate lenti-viruses. Supernatants were centrifuged at $1600 \times g$ for 60 min, and pellets containing lentiviruses were collected.

## Lentivirus transduction

A no-spin inoculation method was used for lentivirus transduction. MEFs at ~90% confluence were dissociated using non-enzymatic Cell-stripper reagent (Corning, #25056CI). For each T75 flask, $5 \times 10^6$ dis-associated cells were incubated with lentiviruses and polybrene (32 µg/mL final concentration) (Sigma-Aldrich, #H9268) in complete culture medium. After incubation at 37 °C overnight, the transduction medium was replaced with fresh medium. At 48 h post-transduction, blasticidin (Invivogen, #ant-bl-1) (5 µg/mL) or puromycin (Invivogen, #ant-pr-1) (4 µg/mL) selection was initiated. For blasticidin selection, the medium was replaced with fresh medium every three days. Sur-viving cells were harvested at 3 and 5 days after puromycin and blas-ticidin selection.

## Establishment of dCas9-VP64 JAM-A$^{-/-}$ x NgR1$^{-/-}$ MEFs and CRISPRa screen

dCas9-VP64-expressing lentivirus was used to transduce JAM-A$^{-/-}$ x NgR1$^{-/-}$ MEFs. After blasticidin selection, monoclonal cells were iso-lated, and dCas9 expression was immunoblotted with Cas9-specific mouse mAb (Biolegend, #844302) (1:1000 dilution). A monoclonal cell line with relatively moderate dCas9 expression was selected and sub-sequently transduced with lentiviruses encoding the CRISPRa A or B sublibraries (Caprano). A lentivirus dose that led to an approximate 30–50% transduction efficiency was used in these experiments. For each sublibrary, $6 \times 10^7$ cells were transduced to guarantee that at least 300 cells were transduced for each sgRNA. Puromycin selection was initiated at 48 h post-transduction, and resistant cells were expanded once for reovirus binding assays. Transduced cells were passaged three times and screened for reovirus binding after each passage. Later passages were anticipated to enhance expression of genes that require longer intervals for transcription. In each reovirus binding assay, the top ~1% of cells that bound reovirus most avidly as determined by fluorescence intensity were sorted into three subpopulations (low, medium, and high binding). sgRNAs in sorted cells were identified by deep sequencing.

## Flow cytometry-based reovirus binding assays

Reovirus virions ($3 \times 10^{12}$ particles) were incubated with 20 µM Alexa Fluor 647 NHS ester (Invitrogen, #A20006) and 50 mM NaHCO$_3$ in a 500 µl total volume rotating at room temperature (RT) for 1.5 h[19]. Unconjugated fluorophore was removed by dialysis overnight in phosphate-buffered saline (PBS) buffer at 4 °C. For reovirus binding assays, cells were dissociated with Cellstripper, incubated with labeled reoviruses ($2 \times 10^5$ virions/cell) at 4 °C for 1 h, and washed three times with ice-cold 2% FBS DMEM. For PirB-specific antibody blockade assay, CHO cells were incubated with PirA/B-specific monoclonal antibody (mAb) 6C1 (BioLegend, #144101) or isotype IgG (BioLegend, #400402) at 4 °C for 1 h and washed twice with PBS prior to reovirus binding. For cell sorting during the CRISPRa screen, unfixed living MEFs were ana-lyzed using a FACSAria II cell sorter (BD Biosciences) operated by FACSDiva™ Software (BD Biosciences, v6.1.3). For other reovirus binding assays, cells were fixed with 1% paraformaldehyde (PFA, Elec-tron Microscopy Sciences) at 4 °C overnight. Reovirus-binding on cells were measured using an LSR II flow cytometer (BD Biosciences) operated by FACSDiva™ Software (v6.1.3) and analyzed by FlowJo software (v10.8.1).

## Deep sequencing and CRISPRa analyses

Genomic DNA of sorted cells was extracted using a DNeasy Blood and Tissue kit (Qiagen, #69504). sgRNA cDNA was amplified as described[91], purified using AMPure XP beads (Beckman Coulter), and assessed for quality using a TapeStation System (Agilent). Samples were sequenced using NexSeq 500 sequencer (Illumina). Deep sequencing Fastq files were mapped against the library reference, and read counts were cal-culated using customized perl scripts and the CaRpools (CRISPR AnalyzeR for Pooled Screens) package in R (version 3.3.2). In the read count ranking of each sample, the top 150–200 candidates were selected to reduce background. To calculate fold change, the relative abundance of selected candidates was compared with the plasmid library. Subcellular distribution of candidate gene products was clas-sified using the Uniprot database. Only proteins with a membrane distribution were included in the analysis for possible validation.

## Detection of reovirus infection by indirect immunofluorescence

CHO cells were transfected with receptor cDNA-expressing plasmids using Transit-LT1. At 48 hpt, cells were adsorbed with reovirus at 37 °C for 1 h. The inoculum was removed and replaced with fresh Ham's F-12 medium supplemented to contain 2% FBS. At 24 hpa, infected cells were detected by indirect immunofluorescence assay (IFA). For antibody-blockade assays, CHO cells were incubated with mAb 6C1 or isotype IgG at 37 °C for 1 h prior to reovirus adsorption. In PirB ectodomain-blockade assays, reovirus was incubated with recombi-nant PirB ectodomain protein (Novus Biologicals, #2754-PB-050) or bovine serum albumin (BSA, New England Biolabs, #B9200S) at RT for 1 h prior to adsorption to PirB-expressing CHO cells at 37 °C for 1 h. In IFA, CHO cells were fixed with ice-cold methanol at −20 °C for 30 min, air dried for 20 min, and incubated with anti-reovirus-specific anti-body. Primary murine cortical neurons were isolated and cultured as described[34,41]. At 7 d post-isolation, primary neurons were adsorbed with T3SA + (MOI of 20 PFU/cell) at 37 °C for 1 h. For antibody-blockade assays, primary neurons were pre-incubated with 5 µg/ml of mAb 6C1 or isotype IgG and adsorbed with T3SA + (MOI of 100 PFU/cell) at 37 °C for 1 h. Infectivity of neurons was quantified by IFA at 24 hpa. Neurons were fixed with 4% PFA at RT for 30 min and washed twice with PBS. Cells were permeabilized with 1% Triton X-100 in PBS at RT for 20 min and blocked with 5% BSA in PBS at RT for 30 min. Reo-virus infection of CHO cells or neurons was detected using rabbit polyclonal reovirus-specific antiserum (1:3000 dilution) diluted in PBS containing 1% BSA[34,41] and Alexa488-conjugated goat rabbit IgG-specific secondary antibody (Invitrogen, #A-11008) (1:500 dilution). Nuclei were stained using DAPI. Immunofluorescent cells were visua-lized and quantified using a Lionheart FX fluorescence microscope (BioTek) operated by Gen5 software (BioTek, v3.12) as described[34,41].

## Binding specificity of reovirus to PirB using model surfaces

Model surfaces (gold-coated silicon) were functionalized with His$_6$-tagged PirB (Abcam, #ab276923) using Ni$^{2+}$-nitrilotriacetate (NTA) chemistry. Surfaces were rinsed with absolute ethanol and dried with nitrogen gas, followed by cleaning for 15 min using a UV-Ozone cleaner (Jetlight). Surfaces were immersed in an ethanol solution containing 0.05 mM NTA-terminated (10%) and polyethylene glycol (PEG)-termi-nated (90%) alkanethiols and left overnight. The following day, the surfaces were rinsed with ethanol and incubated for 1 h in a 40 mM aqueous solution of NiSO$_4$ (pH 7.2). Surfaces were rinsed with water, incubated with His$_6$-tagged PirB (0.1 mg/ml) for 1 h, and rinsed 10 times with virus buffer. Surfaces were used immediately or stored at 4 °C, ensuring that the surfaces remained hydrated at all times.

## Atomic force microscopy tip functionalization

AFM tips (MSCT-D probes for model surfaces and PFQNM-LC-A-CAL for live cells, Bruker) were immersed in chloroform for 10 min, rinsed with ethanol, dried with nitrogen, and cleaned for 15 min in a UV-Ozone cleaner. Tips were placed into a desiccator under Argon with 30 µl of (3-Aminopropyl)triethoxysilane (APTES) and 10 µl of triethylamine (TEA) for 2 h. APTES and TEA were removed, tips were left to cure under Argon for 72 h, and tips were stored under Argon until use. Tips

were functionalized with T3SA- virions or σ3 capsid protein (Cusa Bio, #EP365971) using a heterobifunctional PEG linker as described[20,21]. Cantilevers were washed three times with DMSO and three times with ethanol and dried with nitrogen. NHS-PEG$_{24}$-Ph-aldehyde linker (3.3 mg, Broadpharm) was dissolved in 0.5 ml of chloroform. The ethanolamine-coated cantilevers were immersed in this solution together with 30 µl triethylamine. After 2 h incubation, tips were washed three times with chloroform, dried with nitrogen, and placed on Parafilm (Bemis) in a star formation to orient the cantilevers in the center of the resulting ring. T3SA- virions ($10^9$ particles/ml) or 0.1 mg/mL σ3 protein in a volume of 50 µl was added to the middle of this configuration and incubated with 2 µl of freshly prepared NaCNBH$_3$ solution (6 weight by volume in 0.1 M NaOH [aq]) and incubated at 4 °C for 1 h. The reaction was quenched by adding 5 µl of 1 M ethanolamine (pH = 8) to the solution for 10 min. Tips were washed with virus buffer three times and stored in a 24-well plate in virus buffer at 4 °C for no more than 3 d.

## FD-based AFM on model surfaces

Dynamic force spectroscopy (DFS) experiments were conducted using a ForceRobot 300 (JPK) with same parameters used for the binding probability assays, with a varying retraction velocity of 0.1, 0.2, 1, 5, 10, and 20 µm s$^{-1}$. The results were displayed in DFS plots using Origin software (OriginLab), which also was used to prepare rupture force histograms for distinct LR ranges and apply various force spectroscopy models, as described[20,21]. These models were used to quantify the energy landscape of the interactions and extract the kinetic off-rate ($k_{off}$) and the distance to the transition state x$_u$.

For kinetic on-rate ($k_{on}$) analysis, the BP was determined at a contact time ($t$) in which the tip is in contact with the surface. Those data were fitted and $K_D$ calculated as described[20,21]. The relationship between interaction time ($\tau$) and BP is described by the following equation:

$$BP = A * \left[ 1 - \exp\left( \frac{-(t - t_0)}{\tau} \right) \right] \tag{1}$$

Where $A$ is the maximum BP and $t_0$ the lag time. Origin software was used to fit the data and extract $\tau$. The $k_{on}$ was calculated by the following equation, with $r_{eff}$ the radius of the sphere, $\eta_b$ the number of binding partners, and $N_A$ the Avogadro constant.

$$k_{on} = \frac{\frac{1}{2} * 4\pi r_{eff}^3 * N_A}{3\eta_b \tau} \tag{2}$$

The effective volume in which the interaction can take place corresponds to a half sphere ($4/6\pi r^3_{eff}$), as only within this volume are the molecules grafted onto the tip capable of interacting with their corresponding receptors on the substrate. To assess the statistical significance of the retrieved data, $P$ values were calculated in Origin Pro using Student's t-test.

## Combined FD-based AFM and fluorescence imaging of living cells

Lec2 cells (ATCC, CRL-1736) were cultured in α-Minimal Essential Medium (α-MEM) (GIBCO, # 12571063) supplemented to contain 2 mM L-glutamine, 100 U/ml penicillin and streptomycin, and 10% FBS at 37 °C in 5% CO$_2$. All experiments were conducted using cells at 10–25 passages. Cells were maintained for at least 2 weeks prior to use in experiments. Two days prior to imaging, $10^6$ cells/mL were plated into slide bottom microdishes (Wilco). The day prior to imaging, cells were transfected with a plasmid encoding PirB-2A-GFP containing an auto-cleavable linker using Lipofectamine LTX (Thermo Fisher Scientific, #15338100). These cells were returned to the incubator overnight and rinsed gently with fresh medium three times prior to imaging. In antibody blockade assay, cells expressing PirB or not were incubated with PirB-specific mAb 6C1 at the concentration of 0.1 mg/ml for 30 min.

AFM correlative images of transfected Lec2 cells were acquired using a Bioscope Resolve AFM (Bruker) in PeakForce QNM mode (Nanoscope software v9.2) coupled to an inverted epifluorescence microscope (Zeiss Observer Z.1) or confocal laser scanning microscope (Zeiss LSM 980). All experiments were conducted using a 40x oil objective (NA = 0.95). Cell images (30–50 µm$^2$) were recorded with forces of 500 pN using PFQNM-LC probes (Bruker) having tip lengths of 17 µm, tip radii of 65 nm, and opening angles of 15°. Images of populations of cells were obtained using the optical microscope component to allow for correlative comparisons. All fluorescence and AFM experiments were conducted using cell-culture conditions with the combined AFM and fluorescence microscopy chamber maintained at 37 °C. Cantilevers were calibrated using the thermal noise method, yielding values ranging from 0.08 to 0.14 N m$^{-1}$. The AFM tip was oscillated in a sinusoidal fashion at 0.25 kHz with a 750 nm amplitude. The sample was scanned using a frequency of 0.125 Hz and 128 or 256 pixels per line. Fluorescent images were collected using standard GFP and DIC settings. AFM images and FD curves were analyzed using Nanoscope analysis software (v1.9, Bruker), Origin, and ImageJ (v1.52e). Individual FD curves depicting unbinding events between the cell surface and T3SA- virions were analyzed using Nanoscope analysis and Origin software. The baseline of the retraction curve was corrected using a linear fit on the last 30% of the retraction curve. The loading rate (slope) of each rupture event was determined using the force-time curve. Optical images were analyzed using Zen Blue software (Zeiss GmBH).

## Reovirus RNA quantification

CHO cells were transfected with receptor-encoding cDNAs. At 48 hpt, cells were adsorbed with reovirus T3SA- ($2.5 \times 10^4$ virions/cell) at 37 °C for 1 h. The inoculum was removed and replaced with fresh Ham's F-12 medium supplemented to contain 2% FBS. Cells were lysed at various intervals with lysis buffer of PureLinK RNA Mini kit (Invitrogen, #12183025) for RT-qPCR analysis.

To determine whether SHP-1/2 phosphatases function in reovirus entry, transfected CHO cells were incubated with 50 µM SHP-1/2 inhibitor-NSC-87877 (Sigma-Aldrich, #565851) at 37 °C for 3 h prior and adsorbed with reovirus T3SA- ($2.5 \times 10^4$ virions/cell) in the presence of 50 µM NSC-87877 at 37 °C for 1 h. The inoculum was removed and replaced with fresh Ham's F-12 medium supplemented to contain 2% FBS. Cellular RNA was extracted using a PureLinK RNA Mini kit. Viral S4[49] and cellular β-actin (Applied Biosystems, #Cg04424027) Taqman primer and probe sets were used to amplify cDNA using the qScript XLT 1-Step RT-qPCR ToughMix (Quanta Bio, # 95133-500). cDNA was quantified using a ViiA 7 Real-Time PCR System (Applied Biosystems).

## PirB ICD phosphorylation immuno-detection

CHO cells were transfected with myc-tagged WT or ICD mutant PirB cDNAs. At 48 hpt, CHO cells were incubated with 1 mM Na$_3$VO$_4$ (Sigma-Aldrich, #450243) at 37 °C for 30 min and adsorbed with reovirus T3SA+ ($5 \times 10^5$ virions/cell) in the presence of 1 mM Na$_3$VO$_4$. Cells were lysed at various intervals using ice-cold Pierce IP lysis buffer (Thermo Fisher Scientific, #87787) supplemented with Halt™ Protease and Phosphatase Inhibitor Cocktail (Thermo Fisher Scientific, #78440) and 1 mM Na$_3$VO$_4$. PirB proteins were collected by immunoprecipitation (IP) using myc-specific mouse mAb (Cell Signaling, #2276S) and Dynabead Protein G (Invitrogen, #10004D). Phosphorylation of IP-enriched PirB was detected by immunoblotting as described with modifications[93]. PBS supplemented to contain 1% BSA (Research Products International), 1% PVP-10 (polyvinyl-pyrrolidone) (Sigma-Aldrich, #PVP10), 1% PEG 3500 (Sigma-Aldrich), and 0.2% Tween 20 (Sigma-Aldrich, # P9416) was used to block membranes and dilute antibodies.

Amersham Protran nitrocellulose membranes (Cytiva, #10600033) were incubated in blocking buffer at RT for 1 h and with primary antibody at RT for 1 h. PirB ICD phospho-tyrosines were detected using P-Tyr-1000 MultiMab rabbit mAb (Cell Signaling, #8954) (1:1000 dilution). Total PirB was detected using myc-specific mouse mAb (1:1000 dilution). After washing twice with PBS containing 0.05% Tween-20 (PBST), nitrocellulose membranes were incubated with secondary antibodies including IRDye 680RD goat rabbit IgG-specific IgG (Li-Cor Biosciences, #926-68071) (1:5000 dilution) and IRDye 800CW goat mouse IgG-specific IgG (Li-Cor Biosciences, #926-32210) (1:5000 dilution) at RT 1 h. After washing twice with PBST, membranes were scanned using an Odyssey DLx Imaging system (Li-Cor Biosciences) operated by Image Studio (Li-Cor Biosciences, v5.2). Fluorescence intensity of protein bands was quantified using Image Studio Lite software (Li-Cor Biosciences, v5.2).

### Reovirus uncoating kinetics
CHO cells were transfected with WT or ICD mutant PirB cDNAs. At 48 hpt, cells were incubated on ice for 15 min, adsorbed with reovirus T3SA + ($2 \times 10^5$ virions/cell) on ice for 1 h, washed twice with ice-cold PBS, and incubated at 37 °C. Cells were lysed at various intervals post-adsorption using ice-cold Pierce IP lysis buffer supplemented with Halt™ Protease and Phosphatase Inhibitor Cocktail. Viral proteins in cell lysates were detected by immunoblotting. PBS supplemented to contain 0.05% Tween 20 and 5% non-fat milk (Research Products International) was used to block nitrocellulose membranes and dilute antibodies. Antibody incubation conditions were similar to those used for immuno-detection of phosphorylated tyrosines. Reovirus capsid proteins were detected using rabbit polyclonal reovirus-specific antiserum (1:3000 dilution). Glyceraldehyde 3-phosphate dehydrogenase (GAPDH), used as a loading control, was detected using a GAPDH-specific mouse mAb (Sigma-Aldrich, #CB1001) (1:5000 dilution).

### Animal studies
All mice used in this study were maintained in a specific pathogen-free vivarium at the University of Pittsburgh. Mice were inoculated with reovirus in an animal biosafety level 2 (ABSL2) facility. All mice were maintained at a macroenvironmental temperature range of 68 to 76 °F (20 to 24.4 °C), a relative humidity range of 35% to 55%, and a 12 h/12 h light/dark cycle. Mice of both sexes in equal proportion were used in these experiments, as there is no evidence suggesting that reovirus pathogenesis in newborn mice is influenced by sex.

C57BL/6J x 129S4/SvJaeJ (B6 x 129sv) hybrid mice were used as WT controls due to the hybrid genetic background of PirB$^{-/-}$ mice[38,54]. PirB$^{-/-}$ mice and mice with PirB alleles flanked by loxP sequences (PirB$^{fl/fl}$)[38,54] were provided by Dr. Carla Shatz (Stanford University). PirB$^{fl/fl}$ mice were interbred with mice expressing Cre recombinase under control of a nestin promotor (Jackson Laboratory)[55] to obtain neural-specific PirB-null (NspPirB$^{-/-}$) mice.

For IC inoculations, two-to-four litters of 2-day-old mice (1.5–2.3 g) per genotype were inoculated in the right cerebral hemisphere using a 30-gauge needle and a Hamilton syringe. For PO inoculations, two-to-four litters of 3-day-old mice (2.0 to 3.0 g) per genotype were inoculated using a polyethylene gavage tube and a Hamilton syringe. The titer of virus in the inoculum was confirmed by plaque assay. For assays of viral virulence, inoculated mice were monitored daily for symptoms of disease. Moribund mice or mice with 25% weight loss were euthanized. Morbidity was assessed based on neurological signs including lethargy, seizures, or paralysis. For quantification of viral titers, mouse brains were hemisected along the longitudinal fissure. The right hemisphere was stored in 1 ml PBS for viral titration. The left hemisphere was fixed in 10% neutral-buffered formalin for immunohistochemistry. Other tissues were collected and stored in 1 ml PBS. For

viral titer determination, tissues were frozen and thawed twice and homogenized using a TissueLyser LT (Qiagen). Viral titers were quantified by plaque assay using L929 cells.

### Immunohistology
Mice were euthanized following IC inoculation. Brains were removed and hemisected longitudinally. Right-brain hemispheres were homogenized for viral titer determination. Left hemispheres were fixed using 10% neutral-buffered formalin (NBF) (Thermo Fisher Scientific) for 24 h and submerged into fresh NBF solution. Brain tissues were embedded in paraffin and sliced into 5-mm-thick sections. Tissue-section paraffin was removed by submerging in xylene at RT for 5 min. Tissue sections were then hydrated by serial passage in dilutions of ethanol (100%, 95%, 70%, and 50%) at RT for 5 min and rinsed with distilled water. Reovirus antigen in tissue sections was retrieved by incubating in sodium citrate buffer (10 mM sodium citrate, 0.05% Tween-20, pH = 6) at 95–100 °C for 45 min. For immunofluorescence assays, tissue sections were blocked with 5% BSA in PBS at RT for 1 h and incubated with rabbit polyclonal reovirus-specific antiserum diluted at 1:10,000 ratio in PBS containing 1% BSA for 1 h. After three washes with PBST (0.1% Tween-20, 0.1 M glycine), tissue sections were incubated with 1% BSA in Alexa488-conjugated goat rabbit IgG-specific secondary antibody diluted at 1:500 ratio in PBS and washed three times with PBST (0.1% Tween-20, 0.1 M glycine). Nuclei were stained with DAPI. Tissue sections were mounted with Aqua-Poly/Mount (Polysciences, #18606) overnight at RT and scanned using a Lionheart FX fluorescence microscope.

### Magnetic resonance imaging
Mice for in vivo brain imaging were anesthetized with inhaled isoflurane as described[94,95]. MRI was conducted using a Bruker BioSpec 70/30 USR spectrometer (Bruker BioSpin MRI) operating by ParaVision 5.1 platform at 7-Tesla field strength, equipped with a shielded gradient system and a quadrature radio-frequency volume coil with an inner-diameter of 35 mm. Multi-planar $T_2$-weighted anatomical imaging was acquired using Rapid Imaging with Refocused Echoes pulse sequence with the following parameters: slice number = 15, field of view (FOV) = 1.8 cm, matrix = 256 × 256, slice thickness = 0.65 mm, in-plane resolution = 70 μm, echo time (TE) = 12 ms, RARE factor = 8, effective echo time (TE) = 48 ms, repetition time (TR) = 1551.2 ms, and flip angle (FA) = 180°. MRI data were exported to DICOM format and analyzed by two independent observers blinded to the conditions of the experiment using open-source ITK-SNAP brain segmentation software (http://www.itksnap.org) (version 3.8.0). Regions of inflammation, cerebral hemorrhage, ventricles, and whole brains were manually drawn by observers blinded to the conditions of the experiment based on the Allen mouse brain atlas (https://mouse.brain-map.org/static/atlas) to obtain volumes of each interest region. Inflammation was defined as hyperintensity in the brain tissue, whereas hemorrhage was defined by hypointensity. To account for potentially different brain sizes of PirB$^{fl/fl}$ and NspPirB$^{-/-}$ mice, volumes of each brain region were normalized to the total brain volumes of each individual mouse.

### Statistical analysis
All data except deep sequencing results were analyzed using Graphpad Prism v9.5.1. The number of experimental repeats and statistical tests applied for each assay are provided in the figure legends. Differences in pairwise comparisons were considered to be statistically significant when $P$ values were less than 0.05.

### Reporting summary
Further information on research design is available in the Nature Portfolio Reporting Summary linked to this article.

## Data availability

The data underlying Figs. 1b, 2b–f, 3b, c, 4b–e, h–k, 5, 6, and Supplementary Figs. 3b, 4, 5, 6 are provided as a Source data file and also deposited in Figshare (https://doi.org/10.6084/m9.figshare.22587505). All other relevant data are available from the corresponding authors on reasonable request. Source data are provided with this paper.

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

## Acknowledgements

We are grateful to members of the Dermody laboratory for many useful discussions. We thank the UPMC Children's Hospital of Pittsburgh Flow Cytometry Core for flow cytometry assistance, University of Pittsburgh Health Sciences Sequencing Core for NGS service, and UPMC Children's Hospital of Pittsburgh Animal Imaging Core for MRI analysis. We thank Dr. Carla Shatz from Stanford University for providing mice with genetically modified PirB alleles. This work was supported by the U.S. Public Health Service awards R01 AI174526 (P.S., D.M.S., and T.S.D.) and S10 OD028483, the University of Pittsburgh Center for Research Computing (R.D.), UPMC Children's Hospital of Pittsburgh (P.S. and P.A.), and the Heinz Endowments (T.S.D.). Additional support was provided by the Université Catholique de Louvain, the Fonds National de la Recherche Scientifique (F.R.S.-FNRS), the European Research Council by the European Union's Horizon 2020 research and innovation program (grant number 758224), and the FNRS-Welbio (grant number CR-2019S-01). J.D.S. and M.K. are Postdoctoral Researchers and D.A. is a Research Associate of the FNRS. The funders had no role in study design, data collection and analysis, decision to publish, or preparation of the manuscript.

## Author contributions

P.S. conceived, designed, and conducted experiments, analyzed data, contributed materials and analytic tools, and wrote the paper. J.D.S. conceived, designed, and conducted experiments, analyzed data, and wrote the paper. Y.W., M.K., and D.A. conceived and designed experiments, analyzed data, and contributed materials and analytic tools. G.M.T., D.M.S., O.L.W., P.A., R.D.S.N., K.S., J.J.M., and D.R. conceived, designed, and conducted experiments. A.C.P. contributed materials and analytic tools. T.S.D. conceived and designed experiments, analyzed the data, and wrote the paper. All authors reviewed, critiqued, and provided comments on the manuscript.

## Competing interests

The authors declare no competing interests.
