## [Peer Review File · Nature Communications]

Paired immunoglobulin-like receptor B is an entry receptor for mammalian orthoreovirusREVIEWER COMMENTS

Reviewer #1 (Remarks to the Author):

This manuscript describes the identification of the paired immunoglobulin-like receptor B as an entry receptor of mammalian orthoreovirus in mice. The Mammalian orthoreovirus types 1 and 3 are frequently used to study as model for studies on viral entry, replication and pathogenesis. In mice there is a types specific pathology associated in the brain with reovirus infection. The identification of Pir B as a receptor as further elucidating the mechanisms underlying the type specific pathology.

PirB as an reovirus receptor was identified in in CRISPR activation screen and further evidence comes from a range of studies including single molecule force experiments, binding, entry, uncoating, and infection experiments in cells expressing canonical virus receptors as well as PirB variants, and studies in mice.

The manuscript is very significant to the field. It is coherently written and logically structured. The evidence provided suffices to support the conclusions.

Nevertheless there are a number of issues that may need attention.

1) Entry of reoviruses after binding of their $\sigma 1$ protein to JAM-A depends on a binding of the viral $\lambda 2$ proteins to $\beta 1$ integrins. Is this interaction is also required for entry via the $\sigma 3$ /PirB route? This is an obvious question and the authors' view on this should be covered in the Discussion section of the manuscript.

2) It would be helpful if the authors would outline the purification methods used to generate the reovirus stocks used in this work, as the purity of these preps is essential for following the experimental lines.

3) The labeling of the bars in figure 4D appears erroneous. I assume that the bars labelled with hJAM-A represent the infectivity on the hCAR expressing cells and vice versa. Please check.

4) In the legends to figure 5A it is stated that the T3SA+ viruses were used. Why where the T3SA+ viruses used in this experiment in contrast to other experiments that used T3SA- particles? Please motivate the choice.

Reviewer #2 (Remarks to the Author):

This review of the manuscript by Pengcheng Shang et al. is focused on the AFM binding studies of reovirus to PirB. In this study, the authors provide credible evidence using a variety of techniques and methodologies that PirB is a likely entry point of the orthoreovirus.

The following comments and questions are posed to the authors, especially in relation to the AFM section of the study.

a) The tip-surface interaction in AFM will lead to a variety of force-distance curves and shapes. The authors provide no indication as to what types of force curves were deemed as binding events. Did the authors use a particular distance or a range of distances from the surface for the unbinding event to assume a specific binding event? For example, were there multiple unbinding events during a pull-off and were such curves included or excluded? Or were only single unbinding pull-off events included? More details on which force-distance curves were selected would be helpful. Such a description could be provided in the supplemental section.

b) Is the error provided for $k(\text{on})$, $k(\text{off})$, X_u , and K_d a 95% confidence interval? It would be helpful for the authors to provide the 95% CI for the K_d value reported or state specifically what the error is for the K_d .

c) It is unclear based on the $k(\text{off})$ value provided in line 191, how the authors claim the supramolecular bonds (i.e. the PirB-reovirus bonds) are stable (i.e., this is a kinetic measurement).

d) In the cellular study, did the authors use PirB-specific mAb as a control in addition to the internal controls? The data would be convincing if the authors showed in figure 4I, a decrease in

binding events upon the introduction of PirB-specific mAb. As it is, it is unclear if the entire range of forces observed between 100-350 nm is due to the specific binding of reovirus to PirB. Moreover, the authors deconvolute the probability density curve using specific Gaussian curves. Is it a coincidence that the peak of the Gaussians match the predicted forces in the Williams-Evan model? In other words, how much bias is there in this type of analysis?

e) In the inset in Figure 4c, it is unclear if the fitting model chosen is the best model as only one data point fits the solid line shown by the model. However, it does provide an approximation, which can be confirmed with SPR. The authors should at least cite references that this approach to finding k_{on} provides reasonable values compared to k_{on} obtained from SPR. Do note the units for k_{on} in the graph should be $\mu\text{M}^{-1} \text{s}^{-1}$ and -1 must be written as a superscript.

f) The authors should provide in the supplemental section force distribution plots for figure 4c for each loading rate (i.e., a distribution plot similar to figure 4i for each loading rate).

The AFM study provides strong evidence for binding between reovirus and PirB when using a model surface, but less so with the cell study. The above comments will hopefully strengthen this manuscript.

Reviewer #3 (Remarks to the Author):

The authors use a CRISPRa screen identify the paired Ig-like receptor B (PirB) as a candidate receptor using an engineered immortalized MEF double ko cell line lacking two previously identified reovirus receptors (JAM-A-/-/NgR1-/-) and reovirus strains that fail to bind to sialic acid (a third receptor). They then go on to "validate" PirB by showing that a normally non-permissible cell line (CHO) supports binding by reovirus type 1 and 3 SA- strains after transfection with PirB DNA. Reoviral binding is blocked in a dose-dependent manner by a PirB-specific Mab. Additional studies suggest the PirB D3D4 domains are required for reoviral binding which likely involves the viral sigma-3 protein and occurs with a KD of $\sim 2\text{nM}$. Extending the impact of their studies considerably they go on to show that PirB is involved in the neurotropism and neuropathogenicity of T3 infection.

I found the studies both convincing and comprehensive and the results significantly expand our understanding of reovirus-receptor interactions and biology. The in vivo studies add an important dimension establishing the biological importance of the PirB system in one of the critical experimental model systems (CNS infection) for these viruses. The conclusions drawn are supported by the data provided. I found the methodology sound, state of the art, and often reinforced by the multiplicity of approaches and their integration. The comments below are minor. Their focus is on the in vivo studies as I think this adds a critical dimension to the in vitro work.

Specific comments

- (1) The in vivo studies add an important dimension to the in vitro data establishing PirB as a T3 reo receptor. The authors state that T3SA- (but not T1SA-) virus titers were lower following peroral inoculation in brains of PirB-/- mice compared to WT (PirB+) mice despite similar growth in intestine (Fig 6). The key panel (6a, right) only establishes this is significant at d6 and that by d9 titers are essentially identical. When virus was inoculated intracranially virus grew in PirB-/- mice but less than wild-type. It is not clear (and would be useful to show) whether this difference caught up by d9 (data stops at d6). It would also be useful to know if the tropism of virus was identical in the PirB-/- mice and WT (PirB+) mice. Is this growth in the same cells, but less or due to growth in other (e.g. non-neuronal) cells? Is virus spreading differently (rate or pathways) in the brains of these mice? This could be easily established with immunohistochemistry from sequential and later timepoints (viral titers are generally above typical detection amounts by IHC).
- (2) The survival curves (from neuron-specific PirB-null mice) do suggest some growth in brain (6c) and again are temporally truncated (end at d6) despite the fact that growth is still increasing at this time in other models (panel 6a), and that mortality doesn't really even begin until at least d7 (panel 6d).
- (3) I was surprised by the absence of histology to enable us to see the comparative

neuropathology in either the PirB ^{-/-} versus WT mice and the NspPirB^{-/-} versus PirBfl/fl mice.

(4) MRI data provides an interesting way of evaluating inflammation (and the ability to follow it in living animals). It would help again to see the histological confirmation with basic cell-type staining (e.g. for lymphocytes or to see microglial activation etc.). I might note that simply using "hyperintensity" as equivalent to "inflammation" is very crude at best (edema etc. can induce this too). It might be useful to show some representative MRIs in the supplemental materials.

(5) As a very minor point perhaps correcting titer to tissue weight rather than whole organ might facilitate the intestine versus brain comparison in 6a? (Even just providing some statement about relative sample weights?).

REVIEWER COMMENTS

Reviewer #1 (Remarks to the Author):

This manuscript describes the identification of the paired immunoglobulin-like receptor B as an entry receptor of mammalian orthoreovirus in mice. The Mammalian orthoreovirus types 1 and 3 are frequently used to study as model for studies on viral entry, replication and pathogenesis. In mice there is a type-specific pathology associated in the brain with reovirus infection. The identification of Pir B as a receptor as further elucidating the mechanisms underlying the type-specific pathology.

PirB as a reovirus receptor was identified in in CRISPR activation screen and further evidence comes from a range of studies including single molecule force experiments, binding, entry, uncoating, and infection experiments in cells expressing canonical virus receptors as well as PirB variants, and studies in mice.

The manuscript is very significant to the field. It is coherently written and logically structured. The evidence provided suffices to support the conclusions. Nevertheless, there are a number of issues that may need attention.

1) Entry of reoviruses after binding of their $\sigma 1$ protein to JAM-A depends on a binding of the viral $\lambda 2$ proteins to $\beta 1$ integrins. Is this interaction also required for entry via the $\sigma 3$ /PirB route? This is an obvious question and the authors' view on this should be covered in the Discussion section of the manuscript.

We transfected PirB cDNAs into non-susceptible CHO cells, which do not detectably express $\sigma 1$ receptors. Reoviruses efficiently bind and infect PirB-expressing CHO cells, suggesting that $\sigma 1$ -receptor interactions are not required for interactions with PirB. In unpublished work, we compared internalization into PirB-expressing cells of WT virus and a mutant with disrupted $\lambda 2$ integrin-binding motifs. There was no significant decrease in entry efficiency of the mutant virus, suggesting that $\lambda 2$ -integrin interactions also are not required for $\sigma 3$ -PirB interactions. As mentioned in the Discussion (paragraph 3), we think that the entry of reovirus, especially into neurons, is a step-wise process, in which PirB contributes to viral attachment, internalization, or both entry steps for maximal infection efficiency. However, we do not know whether PirB- $\sigma 3$ binding occurs prior to, concurrently, or following interactions with other outer-capsid proteins and their corresponding receptors.

2) It would be helpful if the authors would outline the purification methods used to generate the reovirus stocks used in this work, as the purity of these preps is essential for following the experimental lines.

The purification method has been described (DOI [dx.doi.org/10.17504/protocols.io.kqqcqvww](https://doi.org/10.17504/protocols.io.kqqcqvww)). We have cited this protocol in the Methods section.

3) The labeling of the bars in figure 4D appears erroneous. I assume that the bars labelled with hJAM-A represent the infectivity on the hCAR expressing cells and vice versa. Please check.

Thank you. This error has been corrected.

4) In the legends to figure 5A it is stated that the T3SA+ viruses were used. Why where the T3SA+ viruses used in this experiment in contrast to other experiments that used T3SA- particles? Please motivate the choice.

We used glycan-blind reoviruses (termed SA-) for most of the assays in this study to eliminate potential non-specific binding to glycosylated cell-surface molecules and allow analysis of the role of PirB in reovirus attachment. However, we used glycan-binding reovirus (termed SA+) to enhance interactions between reovirus and PirB to facilitate detection of transient phosphorylation of intracellular motifs. We included this explanation in the revised manuscript.

Reviewer #2 (Remarks to the Author):

This review of the manuscript by Pengcheng Shang et al. is focused on the AFM binding studies of reovirus to PirB. In this study, the authors provide credible evidence using a variety of techniques and methodologies that PirB is a likely entry point of the orthoreovirus.

The following comments and questions are posed to the authors, especially in relation to the AFM section of the study.

a) The tip-surface interaction in AFM will lead to a variety of force-distance curves and shapes. The authors provide no indication as to what types of force curves were deemed as binding events. Did the authors use a particular distance or a range of distances from the surface for the unbinding event to assume a specific binding event? For example, were there multiple unbinding events during a pull-off and were such curves included or excluded? Or were only single unbinding pull-off events included? More details on which force-distance curves were selected would be helpful. Such a description could be provided in the supplemental section.

To address this point, specific force-distance (FD) curves were selected to show only specific adhesive events based on these criteria: (a) FD curves with extension patterns that could be placed with the worm-like chain model and (b) FD with a rupture distance of > 10 nm, accounting for the extension of the PEG linker. For curves showing multiple peaks, only the last unbinding event was analyzed.

b) Is the error provided for $k(\text{on})$, $k(\text{off})$, X_u , and K_d a 95% confidence interval? It would be helpful for the authors to provide the 95% CI for the K_d value reported

or state specifically what the error is for the K_d .

The error presented is the standard error of the measurement. This description is included in the text to ensure clarity.

c) It is unclear based on the k_{off} value provided in line 191, how the authors claim the supramolecular bonds (i.e. the PirB-reovirus bonds) are stable (i.e., this is a kinetic measurement).

The reovirus-PirB complex displays a k_{off} of $\sim 0.20 \text{ s}^{-1}$. This k_{off} is less than that of the avidin-biotin complex and the His6-Ni-NTA complex (Friddle Proc Natl Acad Sci U S A (2012) DOI: [10.1073/pnas.1202946109](https://doi.org/10.1073/pnas.1202946109)), suggesting that the reovirus-PirB complex is relatively stable.

d) In the cellular study, did the authors use PirB-specific mAb as a control in addition to the internal controls? The data would be convincing if the authors showed in figure 4l, a decrease in binding events upon the introduction of PirB-specific mAb. As it is, it is unclear if the entire range of forces observed between 100-350 nm is due to the specific binding of reovirus to PirB. Moreover, the authors deconvolute the probability density curve using specific Gaussian curves. Is it a coincidence that the peak of the Gaussians match the predicted forces in the Williams-Evan model? In other words, how much bias is there in this type of analysis?

A PirB-specific mAb was used to study reovirus binding to cells. The binding of reovirus to living cells in the presence or absence of PirB mAb is shown in **figure 4h** and **4i**. With respect to the theory of force spectroscopy, by applying the Bell-Evans model (only valid for single-bond rupture), we can extract the kinetic parameters of single bonds (actual fit of the data) and use the predictive Williams-Evans model to evaluate the rupture force of multiple uncorrelated bonds probed in parallel. The force histogram, shown in **figure 4k**, is placed with a multiple Gaussian fit distribution, and the position of the peaks (mean and S.D.) is overlaid onto the histogram. Although this approach is unbiased, since the force peaks in the histogram are quite large and the different fits of the Williams-Evans prediction are close together, good correlation is likely between the Williams-Evans prediction and the maxima of the Gaussian curves. Minimum cutoffs were applied to the fit data. However, peaks determined by the Gaussian fitting algorithm, which requires selecting likely peaks from the histogram, even with no additional supervisory bias, demonstrated that the maxima fall within the range of the points. Therefore, this analysis provides an estimate of the number of bonds established in parallel between virus and its receptors, rather than examining the rupture forces precisely.

e) In the inset in Figure 4c, it is unclear if the fitting model chosen is the best model as only one data point fits the solid line shown by the model. However, it does provide an approximation, which can be confirmed with SPR. The authors should at least cite references that this approach to finding k_{on} provides

reasonable values compared to $k(\text{on})$ obtained from SPR. Do note the units for $k(\text{on})$ in the graph should be $\mu\text{M}^{-1} \text{s}^{-1}$ and -1 must be written as a superscript.

The resolution of the inset in the original **figure 4c** was not optimal, but there are seven different points, and those points align relatively well with the model (see below). This approach is the most commonly used to fit the binding frequency versus contact time data and thus extract the $k(\text{on})$. We recast the inset of **figure 4c** as **figure 4d** for better clarity. As for comparison between AFM and SPR, AFM measurements generally contain less error, as this technique quantifies binding by single molecules, whereas SPR is an ensemble method (Wang Phys Chem Chem Phys (2015) DOI: [10.1039/c4cp03190c](https://doi.org/10.1039/c4cp03190c)). The units have been corrected.

f) The authors should provide in the supplemental section force distribution plots for figure 4c for each loading rate (i.e., a distribution plot similar to figure 4i for each loading rate).

We concur that this analysis will be of value to support the information provided in the main text and have included these data in the Supplementary Information (**supplemental fig. 5**).

Reviewer #3 (Remarks to the Author):

The authors use a CRISPRa screen identify the paired Ig-like receptor B (PirB) as a candidate receptor using an engineered immortalized MEF double ko cell line lacking two previously identified reovirus receptors (JAM-A-/-/NgR1-/-) and reovirus strains that fail to bind to sialic acid (a third receptor). They then go on to “validate” PirB by showing that a normally non-permissible cell line (CHO) supports binding by reovirus type 1 and 3 SA- strains after transfection with PirB DNA. Reoviral binding is blocked in a dose-dependent manner by a PirB-specific Mab. Additional studies suggest the PirB D3D4 domains are required for reoviral binding which likely involves the viral sigma 3 protein and occurs with a KD of $\sim 2\text{nM}$. Extending the impact of their studies considerably they go on to show that PirB is involved in the neurotropism and neuropathogenicity of T3 infection.

I found the studies both convincing and comprehensive and the results significantly expand our understanding of reovirus-receptor interactions and biology. The in vivo studies add an important dimension establishing the biological importance of the PirB system in one of the critical experimental model systems (CNS infection) for these viruses. The conclusions drawn are supported by the data provided. I found the methodology sound, state of the art, and often reinforced by the multiplicity of approaches and their integration. The comments below are minor. Their focus is on the in vivo studies as I think this adds a critical dimension to the in vitro work.

Specific comments:

(1) The in vivo studies add an important dimension to the in vitro data

establishing PirB as a T3 reo receptor. The authors state that T3SA- (but not T1SA-) virus titers were lower following peroral inoculation in brains of PirB^{-/-} mice compared to WT (PirB⁺) mice despite similar growth in intestine (Fig 6). The key panel (6a, right) only establishes this is significant at d6 and that by d9 titers are essentially identical. When virus was inoculated intracranially virus grew in PirB^{-/-} mice but less than wild-type. It is not clear (and would be useful to show) whether this difference caught up by d9 (data stops at d6). It would also be useful to know if the tropism of virus was identical in the PirB^{-/-} mice and WT (PirB⁺) mice. Is this growth in the same cells, but less or due to growth in other (e.g. non-neuronal) cells? Is virus spreading differently (rate or pathways) in the brains of these mice? This could be easily established with immunohistochemistry from sequential and later timepoints (viral titers are generally above typical detection amounts by IHC).

Our data suggest that PirB is not the primary determinant of reovirus tropism in the murine CNS. Instead, we think PirB is required for maximum reovirus infectivity in neurons and may contribute to the efficiency of neuron-to-neuron transmission. Following peroral (PO) inoculation, reovirus disseminates to the CNS using both hematogenous and neural routes. Intracranial (IC) inoculation circumvents the requirement for hematogenous dissemination. Viral loads in the brain are influenced by both the susceptibility of neurons to infection and the route of dissemination. Since PirB is expressed in both the CNS and peripheral nervous system (PNS), we hypothesize that PirB contributes to neural dissemination of reovirus from peripheral sites of infection to the CNS. In PirB^{-/-} mice, reovirus hematogenous dissemination may not be diminished. Therefore, at earlier times after inoculation (6 dpi), reovirus titers are lower in the brain of PirB^{-/-} mice relative to those in WT mice (**fig. 6a**), perhaps reflecting a requirement for PirB in reovirus neural dissemination. However, at later times after inoculation (9 dpi), we think that virus transmitted using hematogenous pathways overcomes the diminished neural dissemination in PirB^{-/-} mice from peripheral tissues to the CNS and neuron-neuron transmission. We included this explanation in the Discussion section (paragraph 5) of the revised manuscript.

We repeated the IC inoculation of WT PirB^{fl/fl} and PirB neural-specific knockout (nspPirB^{-/-}) mice (**fig. 6c**). At 4, 6, and 8 dpi, reovirus titers in the brain of nspPirB^{-/-} mice were significantly less than those in the brain of WT PirB^{fl/fl} mice. We also examined the distribution of reovirus antigen in the brains of WT PirB^{fl/fl} and nspPirB^{-/-} mice at 8 dpi (**supplemental fig. 8**) and found no obvious difference in reovirus tropism. Our lab is currently studying the neural circuits used by reovirus to disseminate in the CNS and testing a function for PirB in neural spread. We plan to publish these results in an independent manuscript.

(2) The survival curves (from neuron-specific PirB-null mice) do suggest some growth in brain (6c) and again are temporally truncated (end at d6) despite the fact that growth is still increasing at this time in other models (panel 6a), and that mortality doesn't really even begin until at least d7 (panel 6d).

As mentioned (Reviewer 3, comment 1), we repeated the IC inoculation of WT PirB^{fl/fl} and nspPirB^{-/-} mice (**fig. 6c**). At 4, 6, and 8 dpi, reovirus titers in the brain of nspPirB^{-/-} mice were significantly less than those in the brain of WT PirB^{fl/fl} mice.

(3) I was surprised by the absence of histology to enable us to see the comparative neuropathology in either the PirB ^{-/-} versus WT mice and the NspPirB^{-/-} versus PirB^{fl/fl} mice.

As mentioned (Reviewer 3, comment 1), we examined the distribution of reovirus antigen in the brains of WT PirB^{fl/fl} and nspPirB^{-/-} mice at 8 dpi (**supplemental fig. 8**) and found no obvious difference in reovirus tropism. This result is concordant with our analysis of reovirus titers in the brains of WT PirB^{fl/fl} and nspPirB^{-/-} mice (**fig. 6b-c**) and infectivity of cultured neurons (**fig. 6f-h**). Our results suggest that PirB is not absolutely required for T3 reovirus replication in CNS, but the receptor contributes to efficient replication at that site.

(4) MRI data provides an interesting way of evaluating inflammation (and the ability to follow it in living animals). It would help again to see the histological confirmation with basic cell-type staining (e.g. for lymphocytes or to see microglial activation etc.). I might note that simply using “hyperintensity” as equivalent to “inflammation” is very crude at best (edema etc. can induce this too). It might be useful to show some representative MRIs in the supplemental materials.

The cell populations that infiltrate the brain following T3 reovirus inoculation and contribute to encephalitis have not been precisely defined. Since MRI is a much more sensitive technique to quantify and globally evaluate inflammation in the brain, we used this approach to visualize T3 reovirus-induced encephalitis. A similar approach was used in a previous study of ours to quantify T1 reovirus-induced hydrocephalus (Stencel-Baerenwald mBio (2015) DOI: [10.1128/mBio.02356-14](https://doi.org/10.1128/mBio.02356-14)). Using MRI to assess encephalitis is the first application in our work on T3 reovirus neuropathogenesis and deepens our understanding of the mechanism of reovirus-induced CNS disease. We have selected representative brain MRI images to compare the extent of inflammation in WT PirB^{fl/fl} and nspPirB^{-/-} mice (**supplemental fig. 7**). As suggested by the reviewer, we clarified in the legend of **figure 6** how inflammation is defined using MRI.

(5) As a very minor point perhaps correcting titer to tissue weight rather than whole organ might facilitate the intestine versus brain comparison in 6a? (Even just providing some statement about relative sample weights?).

There are no significant differences in the weights of intestinal tissue of WT and PirB^{-/-} mice at all times tested (figure below, left panel). We also recalculated viral titers in intestinal tissue based on tissue weight (figure below, middle panel), which shows a trend similar to the viral titers in whole tissues (figure below, right panel). We have indicated in the legend of **figure 6** that the weights of intestinal tissue of WT and PirB^{-/-} mice are comparable at all times tested.

T3 reovirus infects both intestinal epithelia and the enteric nerve system (ENS). Since PirB is expressed in the PNS, we hypothesize that PirB functions in infection of PNS and ENS by neurotropic T3 reovirus and allows T3 reovirus dissemination by neural routes. These points are made in paragraph 5 of the Discussion section.

T3 reovirus replication in the intestine WT and PirB^{-/-} mice. WT and PirB^{-/-} mice were inoculated perorally with 10⁴ PFU of reovirus T3SA-. Mice were euthanized at the times show, and various organs were removed. The weights of intestine tissues were measured (a). Viral titers in intestine were quantified by plaque assay and calculated based on tissue weight (b) or shown as whole tissue viral load (c).

REVIEWERS' COMMENTS

Reviewer #1 (Remarks to the Author):

The additional info provided by the authors further increased the clarity of the manuscript. All my comments have been adequately dealt with.

Reviewer #2 (Remarks to the Author):

The manuscript authors have fully addressed the concerns of this reviewer and provided the requested information in the supplemental section. The revisions made by the authors provide clarity to the manuscript. This is an excellent paper and worthy of publication in Nature Communications.

Reviewer #3 (Remarks to the Author):

I believe the authors have thoughtfully and appropriately responded to my comments and queries.